# Balanced Hyperbolic Embeddings Are Natural Out-of-Distribution Detectors

## Abstract

Out-of-distribution recognition forms an important and well-studied problem in deep learning, with the goal to filter out samples that do not belong to the distribution on which a network has been trained. The conclusion of this paper is simple: a good hierarchical hyperbolic embedding is preferred for discriminating in- and out-of-distribution samples. We introduce Balanced Hyperbolic Learning. We outline a hyperbolic class embedding algorithm that jointly optimizes for hierarchical distortion and balancing between shallow and wide subhierarchies. We can then use the class embeddings as hyperbolic prototypes for classification on in-distribution data. We outline how existing out-of-distribution scoring functions can be generalized to operate with hyperbolic prototypes. Empirical evaluations across 13 datasets and 13 scoring functions show that our hyperbolic embeddings outperform existing out-of-distribution approaches when trained on the same data with the same backbones. We also show that our hyperbolic embeddings outperform other hyperbolic approaches, can beat state-of-the-art contrastive methods, and natively enable hierarchical out-of-distribution generalization.

## 1 Introduction

Detecting out-of-distribution samples is crucial in real-world settings to make classification predictions reliable and ensure a safe deployment of trained models (Liu et al., 2021). These models are typically trained on datasets with closed-world assumptions He et al. (2015), referred to as in-distribution (ID) data, and testing samples that significantly deviate from training distribution are referred to as out-of-distribution (OOD) data. A wide range of works have proposed approaches to score the likelihood of a testing sample being OOD or not (Yang et al., 2022; Zhang et al., 2023b). Since OOD samples are unseen during training, the key approaches to determine OOD score for a model are based only on ID samples. Scoring functions to classify OOD samples are primarily based on model's confidence (Hendrycks & Gimpel, 2016; Liang et al., 2018; Hendrycks et al., 2022; Liu et al., 2020b) or the feature distance from ID embeddings (Lee et al., 2018b; Sun et al., 2022)

Recent literature has highlighted that scoring functions and optional training or outlier exposure are not the only considerations for effective out-of-distribution detection; the choice of embedding space directly influences out-of-distribution discrimination (Ming et al., 2023; Lu et al., 2024). In this paper, we find that hyperbolic embeddings naturally help to discriminate in- and out-of-distribution samples. We show this in Figure 1a. Different from the Euclidean classifier, the hyperbolic classifier provides strongly uniform distributions for samples near the origin and strongly peaked distributions for samples near the boundary. This observation matches directly with recent literature on hyperbolic learning (Mettes et al., 2023). Hyperbolic geometry makes it possible to deal with hierarchical distributions (Nickel & Kiela, 2017), spatial object boundaries (Ghadimi Atigh et al., 2022), adversarial shifts (Guo et al., 2022), and uncertainty (Franco et al., 2023). All papers find a direct link between the norm of representations in hyperbolic space and sample certainty, akin to Figure 1a. We seek to take advantage of this natural property in hyperbolic learning to help discriminate out-of-distribution from in-distribution samples.

This paper introduces Balanced Hyperbolic Learning. We first represent classes as prototypes in hyperbolic space based on their hierarchical relations. This naturally leads to a desirable ordering, where in-distribution classes end up near the edge of the Poincaré ball and less specific (i.e. more general and uncertain) inner nodes end up closer to the origin as a function of their hierarchical

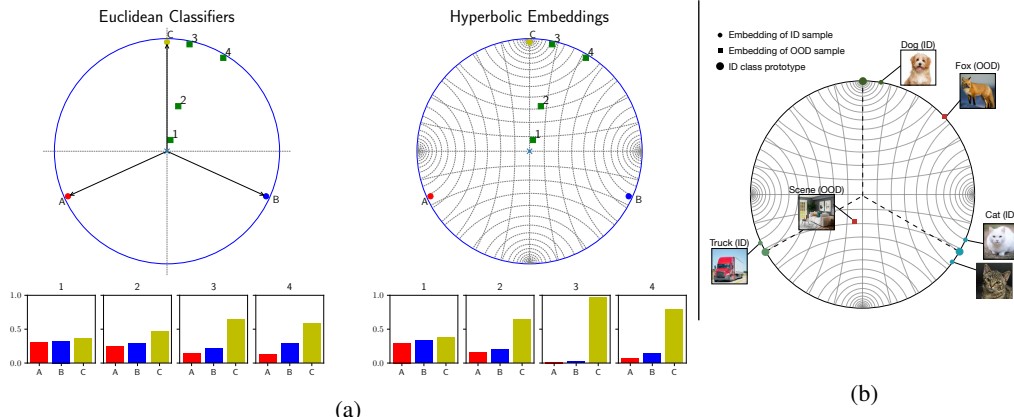

Figure 1: **(a)Examining distances in different embedding spaces.** [Top] The ● represents classifiers in Euclidean space (left) and prototypes in hyperbolic space (right, here a Poincaré disk). The ■ represents image embeddings for various images. In Euclidean space, logits are obtained by the dot product with classifiers, while in the proposed hyperbolic method, logits are based on the distance to the class prototype, measured along the geodesic. [Bottom] shows how the softmax distribution of the image embeddings changes based on the distance to the classifier. In hyperbolic space, the model gives higher confidence to images near the classification boundary and relatively lower confidence to those further away, which is a desirable property for detecting out-of-distribution samples. **(b) Illustration of desirable hyperbolic embeddings for OOD detection.** Depending on relation to ID samples, OOD samples lie between ID clusters (slightly related) or closer to the origin (unrelated).

depth. We find that existing hyperbolic embedding methods are biased towards deeper and wider sub-trees, with smaller sub-trees pushed towards the origin. This is in direct conflict with Figure 1a, since it leads to less uniform softmax distributions for OOD samples that end up near the origin. We propose a distortion-based loss function with norm balancing across all hierarchical levels to obtain class embeddings and optimize ID samples to align with their class prototypes. Over the years, many scoring functions have been introduced in out-of-distribution literature. Rather than introduce yet another alternative, we show how existing functions effortlessly generalize to work with prototypes in hyperbolic space. Figure 1b illustrates the outcome, where OOD samples lie between ID clusters or near the origin. Empirical results on a wide range of datasets and scoring functions show that our hyperbolic embeddings structurally lead to better OOD discrimination.

## 2 PRELIMINARIES

### 2.1 OUT-OF-DISTRIBUTION DETECTION

Let $\mathcal{X} := \mathbb{R}^n$ and $\mathcal{Y}^{in} := \{1, ..., C\}$ denote the input and label space of the in-distribution training data for multi-class image classification. For this closed-world setting, the data $D_{id} = \{(\mathbf{x}_i, y_i)\}_{i=1}^{N}$ is drawn *i.i.d* from $\mathcal{P}_{\mathcal{X}\mathcal{Y}^{in}}$ and assumes the same distribution during training and testing. The aim of Out-of-Distribution (OOD) detection is to decide whether a sample $\mathbf{x} \in \mathcal{X}$ is from $\mathcal{P}_{\mathcal{X}}$ (ID) or not (OOD). We consider the canonical OOD setting (Hendrycks & Gimpel, 2016) where OOD samples are from unknown classes, *i.e.* $\mathcal{Y}^{id} \bigcap \mathcal{Y}^{ood} = \emptyset$. With $S(\mathbf{x})$, a scoring function on logits or features of a trained model, an input $\mathbf{x}$ is identified as OOD if $S(\mathbf{x}) < \sigma$, where threshold $\sigma$ is a level set parameter determined by the false ID detection rate (e.g., 0.05) (Ming et al., 2022; Chen et al., 2017).

### 2.2 THE POINCARÉ BALL MODEL OF HYPERBOLIC SPACE

This paper works with the most commonly used model of hyperbolic geometry in deep learning, namely the Poincaré ball model (Khrulkov et al., 2020; Ghadimi Atigh et al., 2021; van Spengler et al., 2023). The $d$-dimensional Poincaré ball with constant negative curvature $-c$ is defined as the Riemannian manifold $(\mathbb{B}_c^d, \mathfrak{g}_c)$, where $\mathbb{B}_c^d = \{\mathbf{x} \in \mathbb{R}^d : \|\mathbf{x}\|^2 < 1/c\}$, equipped with the Riemannian metric tensor (Cannon et al., 1997),

$$\mathfrak{g}_c = \lambda_{\mathbf{x}}^c \mathfrak{g}^E, \quad \lambda_x^c = \frac{2}{1 - c \|\mathbf{x}\|^2}, \tag{1}$$

where $\mathfrak{g}^E = I_d$ denotes the Euclidean metric tensor. The Euclidean metric is changed by a simple scalar field, hence the model is conformal (i.e. angle preserving), yet distorts distances.

**Definition 2.1** (Induced distance and norm). The induced distance between two points $\mathbf{x}, \mathbf{y}$ on the Poincaré ball $\mathbb{B}_c^d$, is given by $d_c(\mathbf{x}, \mathbf{y}) = (2/\sqrt{c}) \tanh^{-1}(\sqrt{c} \|-\mathbf{x} \oplus_c \mathbf{y}\|)$. For the Poincaré ball with $c = -1$, the induced distances becomes,

$$d_{\mathbb{B}}(\mathbf{x}, \mathbf{y}) = \cosh^{-1}\left(1 + 2\frac{\|\mathbf{x} - \mathbf{y}\|^2}{(1 - \|\mathbf{x}\|^2)(1 - \|\mathbf{y}\|^2)}\right). \tag{2}$$

The Poincaré norm is then defined as:

$$\|\mathbf{x}\|_{\mathbb{B}} := d_{\mathbb{B}}(0, \mathbf{x}) = 2 \tanh^{-1}(\|\mathbf{x}\|). \tag{3}$$

**Definition 2.2** (Exponential map). The exponential map provides a way to map a vector from the tangent spaces onto the manifold, $\mathcal{T}_x\mathbb{R}^d \to \mathbb{B}_c^d$, given by (Ganea et al., 2018):

$$\exp_{\mathbf{v}}(\mathbf{x}) := \mathbf{v} \oplus_c \left(\tanh\left(\sqrt{c}\frac{\lambda_{\mathbf{x}}^c \|\mathbf{x}\|}{2}\right) \frac{\mathbf{x}}{\sqrt{c} \|\mathbf{x}\|}\right), \tag{4}$$

where $\mathbf{x} \in \mathbb{B}^d$ and $\mathbf{v} \in \mathcal{T}_x\mathbb{R}^d$ with $\oplus_c$, the Möbius addition (Ungar, 2022):

$$\mathbf{v} \oplus_c \mathbf{w} = \frac{(1 + 2c \langle \mathbf{v}, \mathbf{w} \rangle + c \|\mathbf{w}\|^2)\mathbf{v} + (1 - c \|\mathbf{v}\|^2)\mathbf{w}}{1 + 2c \langle \mathbf{v}, \mathbf{w} \rangle + c^2 \|\mathbf{v}\|^2 \|\mathbf{w}\|^2}. \tag{5}$$

In practice, $\mathbf{v}$ is set to the origin, which simplifies the exponential map to

$$\exp_0(\mathbf{x}) = \tanh(\sqrt{c} \|\mathbf{x}\|) \frac{\mathbf{x}}{\sqrt{c} \|\mathbf{x}\|}. \tag{6}$$

## 3 METHOD

### 3.1 OVERVIEW OF THE PROPOSED METHOD

The hypothesis of this paper is that hyperbolic embeddings, accompanied by a hierarchical organization of in-distribution classes, are a natural match for out-of-distribution detection. The in-distribution hierarchy is given as $G = (V, E)$ with $|V| > C$ denoting the $C$ classes as leaf nodes with additional inner nodes leading to a root node. While an additional assumption, we find that such hierarchical information typically comes for free, for example by using large-scale knowledge graphs such as WordNet (Miller, 1995) or VerbNet (Schuler, 2005), or simply by prompting a large language model to provide a hierarchical decomposition of a set of classes (Liu et al., 2024).

The proposed method consists of two steps, (i) we first learn *balanced hyperbolic embeddings* for class labels in the hyperbolic space, $\mathbb{B}^d$, by optimizing for pairwise distances between class labels in the hyperbolic space to be equivalent to the graph distance defined by a given hierarchy of the classes. (ii) We then learn a network encoder $f_\theta : \mathcal{X} \to \mathbb{R}^d$ and project the embeddings to the hyperbolic space, $\mathbb{B}^d$, with an exponential map. A distance-based loss between image features and class labels as prototypes in the hyperbolic space is used to shape the embedding space and enable the learning of $f_\theta$, which will produce naturally discriminative embeddings for OOD detection. We then show that we can use our resulting model with the plethora of existing scoring functions to determine OOD scores.

### 3.2 BALANCED HYPERBOLIC EMBEDDING AND LEARNING

Given a hierarchy represented as a directed graph $G = (V, E)$ with $n$ nodes, we compute pairwise graph distances between all nodes by Dijkstra's algorithm for the undirected graph, represented as $d_{ij} = d_G(v_i, v_j)$ where $v_i, v_j \in V$. We initialize the hyperbolic embeddings corresponding to the $n$ graph nodes as $P_{\mathbb{B}} = \{p_1, p_2, .., p_n\}$ where $p_i, p_j \in \mathbb{B}_c^d$. Our objective for Balanced Hyperbolic

---

**Algorithm 1** Obtaining Balanced Hyperbolic Embeddings

---

**Input:** Poincare ball $\mathbb{B}_c^d$ with $c = -1$ and $d = 64$, hierarchy $G = (V, E)$,
graph distance matrix $d_G$, total epochs $e$
**Output:** Balanced Hyperbolic Embeddings, $P_\mathbb{B}$

$P_\mathbb{B}^0 = \text{PoincaréEmbeddings}(G)$                                                     Initialization
**for** $i$ in $e$ **do**;
    $L_d = \sum_{i,j}(d_\mathbb{B}(p_i, p_j) - d_G(v_i, v_j))/d_G(v_i, v_j)$              Distortion loss, Equation 7
    $L_n = 1/n \sum_l \sum_{n^l}(p_i^l - m^l)$                            Norm loss, Equation 9
    $L = L_d + i/e \cdot \tau \cdot L_n$
    $P_\mathbb{B}^i = \mathfrak{R}_{P_\mathbb{B}^{i-1}}(-\eta_i \Delta_R L(P_\mathbb{B}^{i-1}))$               Riemannian gradient update
**end for**

---

Embeddings is to optimize embeddings $P_\mathbb{B}$, such that the distances between any two nodes, $(p_i, p_j)$ is similar to distances between the graph nodes, $(v_i, v_j)$. We do so by directly minimizing the distortion Sala et al. (2018) between the hyperbolic and graph distances. Additionally, we want to avoid a bias towards broad sub-trees by balancing the hyperbolic norms of nodes at the same level of granularity. An overview is provided in Algorithm 1, below we outline our losses in detail.

**Distortion loss.** We first initialize $P_\mathbb{B}$ using the Poincaré Embeddings of Nickel & Kiela (2017) to obtain coarsely aligned embeddings. We want to optimize the embeddings such that their pairwise distances, given by Equation 2, closely reflect the graph's hierarchical distances $d_{ij}$, with minimal error. We do so by directly optimizing this difference:

$$L_d = \frac{d_\mathbb{B}(p_i, p_j) - d_G(v_i, v_j)}{d_G(v_i, v_j)}. \tag{7}$$

**Norm loss.** Ideally, nodes on the same level in the hierarchy should have the same norm, ensuring a uniform distribution across levels. However, this uniformity often doesn't hold in current algorithms. It is especially evident in imbalanced graphs where one of the paths might have fewer leaf nodes, leading to uneven embeddings (refer Appendix A). We introduce an additional norm-based constraint to promote a more balanced and representative embedding of the hierarchical structure within the Poincaré ball. We want all points within a particular level, $l$ of the hierarchy, to have the same norm (eq. 3). This is done by ensuring the norm of each point, $p_i^l$ in level $l$ is close to the average norm. The average norm for level $l$ is calculated as

$$m^l = \frac{1}{n^l} \sum_1^{n^l} \|p_i^l\|_\mathbb{B}, \tag{8}$$

where $n^l$ is the number of points at level $l$. The overall norm loss is given as a sum over all nodes with respect to the mean at their hierarchical level:

$$L_n = \frac{1}{n} \sum_l \sum_{n^l} (p_i^l - m^l). \tag{9}$$

As shown in Algorithm 1, we initialize a Poincaré ball model with curvature $c = -1$ and obtain coarse embeddings with Poincaré Embeddings trained for 100 epochs. The inputs for the training are the edges and the targets are the pairwise distances $d_{ij}$. We train the model with the joint loss from $L_d$ and $L_n$ with Riemannian SGD (Becigneul & Ganea, 2018) for 10,000 epochs. We increase the contribution of the norm loss to the total loss as a function of the number of epochs. The multiplying factor, $\tau$, for the norm loss depends on the depth of the hierarchy. We empirically find that $\tau$ can be set to $0.01$ for two-level hierarchies and $0.1$ for any deeper hierarchy. We set the dimension of the Poincaré ball $\mathbb{B}_c^d$ to 64, following the literature (Khrulkov et al., 2020).

**Learning ID data with balanced hyperbolic embeddings.** During training, we project input images to the same space as the hyperbolic embeddings, such that we can optimize their alignment. We can obtain a hyperbolic representation of an input image $\mathbf{x}$ using equation 6 as follows:

$$\mathbf{z} = \exp_0^c(\mathcal{F}(\mathbf{x}; \theta)), \tag{10}$$

where $\mathcal{F}_\theta(\mathbf{x}) \in \mathbb{R}^d$ denotes an arbitrary network backbone that yields a $d$-dimensional Euclidean output representation for each input image $\mathbf{x}$.

With classes given as prototypes from $P_\mathbb{B}$ and images as vectors $\mathbf{z}$ in the same hyperbolic space, we keep the prototype fixed and define a hyperbolic distance-based cross-entropy objective, akin to Long et al. (2020), where $d_\mathbb{B}$ is the geodesic distance defined in equation 2:

$$L = -\frac{1}{N} \sum_{n=1}^{N} \sum_{k=1}^{C} \log \frac{\exp(-d_\mathbb{B}(\mathbf{z}_{(n,k)}, p_k))}{\sum_{i=1}^{C} \exp(-d_\mathbb{B}(\mathbf{z}_{(n,i)}, p_i))}, \tag{11}$$

### 3.3 Hyperbolic out-of-distribution scoring

Scoring functions have been well-studied in out-of-distribution detection. We believe that adding yet another does not fully hammer down our point that hyperbolic embeddings are powerful for out-of-distribution detection in the broad sense. We will therefore focus on generalizing a wide range of existing functions to operate on hyperbolic embeddings or prototypes. As we will show, this requires minimal to no changes. We exclude functions that use additional outlier data, as our goal is to show the effect of hyperbolic embeddings as is. We also exclude Mahalanobis-based functions, as each explicitly assume features to be Euclidean. We perform evaluations on 13 different scoring functions in total: MSP (Hendrycks & Gimpel, 2016), Temperature Scaling (Guo et al., 2017), ODIN(Liang et al., 2018), Energy(Liu et al., 2020b), Activation Shaping(ASH) (Djurisic et al., 2022), Generalized Entropy (GEN) (Liu et al., 2023) use logits to design their OOD score. Gram (Sastry & Oore, 2020), KNN (Sun et al., 2022), DICE (Sun & Li, 2022), RankFeat (Song et al., 2022), SHE(Zhang et al., 2022b), NNGuide (Park et al., 2023) and SCALE (Xu et al., 2023). All functions use features, logits, or probabilities at the intermediate or last layer.

MSP and Temp Scaling take the maximum of the softmax of the logits, $f_i$ as the score, and ODIN additionally adds a noise perturbation to the input. This is directly applicable in our setup as well, with the only difference that the logits are now given by the negative of hyperbolic distances, $-d_\mathbb{B}(\mathbf{z}_i, p_i)$ for the hyperbolic embedding of $\mathbf{z}_i$ of image $x_i$ and class prototype $p_i$ The energy score is defined as $E(\mathbf{x}, f) = -T \cdot \log \sum_i^C e^{f_i(\mathbf{x})/T}$ where $f_i$ is the logit corresponding to $i$-th label and $T$ is the temperature hyperparameter. In our method, with $\mathbf{z} = \exp_0^c(f_i(\mathbf{x}_i))$, this score is given by

$$E(\mathbf{x}, f) = T \cdot \log \sum_i^C e^{-d_\mathbb{B}(\mathbf{z}_i, p_i)/T}. \tag{12}$$

Note that we no longer take the negative energy values because our logits are already given by the negative of the prototype distance. Throughout the experiments, we use a $T = 10$ in the energy-based scoring function for ours and $T = 1$ for the baseline, as these are the best performing settings for both. All other scoring functions use features at the intermediate or last layer. We have investigated generalizing these functions to operate the exponential mapping and found no clear difference. Therefore, for scoring functions using features or intermediate layers, we compute scores on the euclidean features in our approach as well for direct comparison to Euclidean-trained counterparts. We note that the features have in our case been optimized to align with hyperbolic class prototypes, hence these features still benefit from our approach.

## 4 Experimental setup

**Datasets.** For a standard out-of-distribution detection setting, we follow the OpenOOD benchmark (Yang et al., 2022; Zhang et al., 2023b). Our in-distribution datasets are CIFAR-100 (Krizhevsky et al., 2009) and Imagenet-100 (Deng et al., 2009). For *CIFAR-100*, we use CIFAR-10 (Krizhevsky et al., 2009) and TinyImagenet (Le & Yang, 2015) as near out-of-distribution datasets. MNIST (Deng, 2012), Textures(Cimpoi et al., 2014), SVHN (Yuval, 2011) and Places365(Zhou et al., 2017) serve as far out-of-distribution datasets. For *Imagenet-100*, SSB-hard (Vaze et al., 2021) and NINCO (Bitterwolf et al., 2023) are near out-of-distribution data, with iNaturalist(Van Horn et al., 2018), Textures(Cimpoi et al., 2014), and OpenImage-O (Wang et al., 2022) as far out-of-distribution data. For all evaluations, we only assume hierarchical information for the in-distribution classes, nothing is assumed for the out-of-distribution data. For the core evaluations, we follow the OpenOOD protocol (Zhang et al., 2023b). As an extra verification, we report

Table 1: **Balanced Hyperbolic Learning across 13 scoring functions** evaluated on OpenOOD with CIFAR-100. We find that scoring functions benefit from relying on hyperbolic embeddings as the final layer, especially for lowering false positive rates.

| | FPR@95 ↓ | | AUROC ↑ | | AUPR ↑ | | n-AUROC ↑ | |
| --- | --- | --- | --- | --- | --- | --- | --- | --- |
| | Base | Ours | Base | Ours | Base | Ours | Base | Ours |
| **MSP** (Hendrycks & Gimpel, 2016) | 58.24 | **49.46** | 77.05 | **82.43** | 64.37 | **70.41** | 77.48 | **78.01** |
| **TempScale** (Guo et al., 2017) | 57.54 | **48.61** | 78.18 | **83.02** | 64.73 | **71.13** | 78.29 | **78.25** |
| **Odin** (Liang et al., 2018) | 60.96 | **49.45** | 76.63 | **82.96** | 62.49 | **70.26** | 78.06 | 77.94 |
| **Gram** (Sastry & Oore, 2020) | 83.33 | **57.78** | 62.31 | **76.84** | 43.58 | **64.64** | 46.60 | **62.37** |
| **Energy** (Liu et al., 2020b) | 58.47 | **55.41** | 77.65 | **81.74** | 64.30 | 61.83 | 78.18 | 77.45 |
| **KNN** (Sun et al., 2022) | 47.95 | **44.00** | 83.29 | **85.50** | 71.02 | **73.71** | 78.45 | **78.84** |
| **DICE** (Sun & Li, 2022) | 64.61 | **54.67** | 74.35 | **80.96** | 59.43 | **66.35** | 74.29 | **77.64** |
| **Rank Feat** (Song et al., 2022) | 73.03 | **49.91** | 68.98 | **81.25** | 51.89 | **68.51** | 60.59 | **64.87** |
| **ASH** (Djurisic et al., 2022) | 67.48 | **55.29** | 76.88 | **76.83** | 57.43 | **64.89** | 75.20 | **75.44** |
| **SHE** (Zhang et al., 2022b) | 77.07 | **53.78** | 67.09 | **82.02** | 49.58 | **67.21** | 68.76 | **78.77** |
| **GEN** (Liu et al., 2023) | 54.66 | **48.70** | 79.21 | **82.96** | 67.25 | **70.98** | 79.08 | 78.18 |
| **NNGuide** (Park et al., 2023) | 65.44 | **57.93** | 76.37 | **81.23** | 60.56 | **63.13** | 75.27 | 77.47 |
| **SCALE** (Xu et al., 2023) | 57.65 | **53.31** | 79.68 | 79.20 | 67.88 | **66.71** | 77.66 | **77.18** |

the performance on the benchmark datasets defined by Hendrycks and Gimpel (Hendrycks & Gimpel, 2016). We are also interested in hierarchical out-of-distribution evaluations. For this, we use the CIFAR-100 OSR splits from OpenOOD (Zhang et al., 2023b) for in- and out-of-distribution and generate hierarchies and balanced hyperbolic embeddings only for the in-distribution classes.

CIFAR100 has a two-level hierarchy with superclasses and classes as defined by the dataset itself. For CIFAR-100 OSR splits from OpenOOD (Zhang et al., 2023b), we use only part of the hierarchy corresponding to the split, leading to imbalanced hierarchies. For ImageNet100, we use the pruned 6-level hierarchy and split from Linderman et al. (2023).

**Implementation details.** For CIFAR-100 and ImageNet-100, we train a ResNet-34 for 200 epochs. The batch size is 128 for CIFAR and 256 for ImageNet. We use SGD with 0.9 momentum and a learning rate of 0.1 with cosine annealing scheduler (Loshchilov & Hutter, 2016), with a weight decay of 0.0005. We perform 3 independent training runs for each method and report the average performance. For a fair comparison to other hyperbolic methods, we use the same setting as our method whenever possible. The hyperbolic prototypes are scaled by a factor 0.95 for a more stable training, and the resulting logit distances are multiplied by a temperature factor $\gamma = 10$.

**Evaluation metrics.** Following OpenOODv1.5 (Zhang et al., 2023b), we use the AUROC, AUPR and FPR@95 scores as metrics. We also report near- and far-OOD AUROC averaged over all out-of-distribution datasets in each group. In the hierarchical evaluations, we report out-of-distribution metrics on CIFAR-OOD along with the benchmark datasets. We are also interested in measuring whether out-of-distribution samples conform to the hierarchical structure of the in-distribution data, without any knowledge of the out-of-distribution classes during training. We report two hierarchical metrics: hierarchical distance@k (Bertinetto et al., 2020) on in- and out-of-distribution samples and the hierarchical similarity index (Dengxiong & Kong, 2023).

# 5 EXPERIMENTAL RESULTS

We evaluate our method for OOD detection, benchmarking it against a baseline Euclidean network across 13 scoring functions on various ID and OOD datasets. Additionally, we ablate the effects of distortion and balancing, compare it with other hyperbolic approaches, and state-of-the-art OOD methods. Finally, we provide a brief overview of the hierarchical OOD setting, with additional analysis and details presented in Appendix C.

**Out-of-distribution comparison overview.** In the first experiment, we focus on a thorough comparative evaluation of Balanced Hyperbolic Learning compared to the standard in out-of-distribution detection with a softmax cross-entropy classifier. The purpose of the experiment is to evaluate how well a wide range of existing out-of-distribution scoring functions work when making the switch from a standard classification head to our hyperbolic embeddings. For this experiment, we compare

Table 2: **Balanced Hyperbolic Learning across 5 scoring functions** evaluated on OpenOOD with ImageNet100. Our approach is also viable with ImageNet classes as in-distribution data.

| | FPR@95 ↓ | | AUROC ↑ | | AUPR ↑ | | n-AUROC ↑ | |
|---|---|---|---|---|---|---|---|---|
| | Base | Ours | Base | Ours | Base | Ours | Base | Ours |
| **MSP** Hendrycks & Gimpel (2016) | 49.08 | **47.98** | 90.06 | **91.46** | 89.10 | **92.89** | 84.56 | **86.00** |
| **Odin** Liang et al. (2018) | 42.13 | **39.79** | 91.31 | **93.42** | 90.23 | **94.40** | 80.24 | **85.29** |
| **Gram** Sastry & Oore (2020) | 83.46 | **63.25** | 72.18 | **80.28** | 74.40 | **88.60** | 63.63 | **81.13** |
| **Energy** Liu et al. (2020b) | 45.23 | **39.38** | 92.03 | **93.49** | 91.36 | **94.27** | 82.58 | **87.18** |
| **KNN** Sun et al. (2022) | **37.13** | 45.74 | **93.58** | 92.99 | 94.33 | **98.81** | 81.11 | **87.86** |

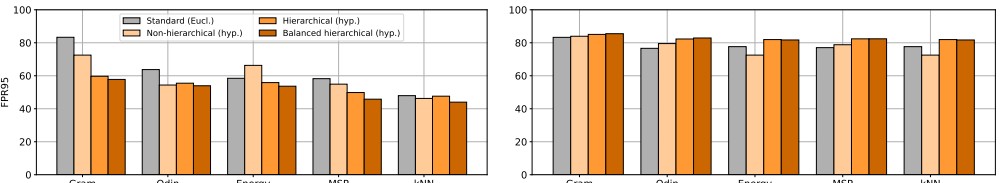

Figure 2: **Out-of-distribution ablation study.** Across scoring functions and evaluation metrics, we find that hyperbolic embeddings in combination with a distortion-based objective and subhierarchy balancing all help to get the best out-of-distribution scores. The ID data is CIFAR-100. FPR@95 ↓ (left) and AUROC ↑ (right).

the baseline to ours across all datasets for FPR@95, AUROC, AUPR, and near-AUROC. For the baseline and ours, we use the exact same backbone and training procedure.

The results of the comparison with OpenOOD for CIFAR-100 are shown in Table 1. Each number represents the performance averaged across all in- and out-of-distribution datasets. We find that our hyperbolic embeddings have a positive effect on all 13 scoring functions. Despite the unique nature of many scoring functions, ranging from density-based to perturbation-based approaches, they all benefit from relying on hyperbolic embeddings to perform the out-of-distribution detection. Interestingly, some scoring functions which are less effective in standard out-of-distribution detectors become highly viable functions on top of hyperbolic embeddings. As example, the canonical maximum softmax probability function yields an improvement from 58.24 to 49.46 in terms of FPR@95.

In Table 2, we show the results with ImageNet100 as in-distribution dataset, with the same outcome. We conclude that Balanced Hyperbolic Learning enriches existing scoring functions without the need for any more parameters or longer training/testing time.

**Effect of distortion and balancing.** The strong out-of-distribution performance of our approach is a result of using hyperbolic embeddings with hierarchical distortion and subhierarchy balancing. To understand which aspect is most crucial for the final performance, we have performed an ablation study to dissect these aspects. We use five well-known scoring functions. For each, we train a standard (Euclidean) baseline. We also train a model that uses hyperbolic embeddings without hierarchies by taking one-hot vectors as class prototypes, scaled down by a factor $0.95$ to fit inside the Poincaré ball. We also train our distortion-based hierarchical embeddings with and without balancing. In Figure 2, we compare all four variants for both the FPR@95 and the AUROC metrics. Across all scoring functions, we observe a similar trend, where each addition improves the results. We first notice that simply using one-hot prototypes in hyperbolic space already for 4/5 (FPR@95) and 3/5 (AUROC) scoring functions. Including our distortion-based hierarchical objective and balancing on top continue to improve the results. We conclude that balancing, distortion, and hyperbolic embedding all matter for out-of-distribution detection.

**Comparison to Hierarchical Embedding Methods.** Several hyperbolic embeddings have previously been proposed for embedding hierarchical knowledge, with Poincaré Embeddings Nickel & Kiela (2017) and Hyperbolic Entailment Cones Ganea et al. (2018) as the most popular algorithms. In the third experiment, we investigate whether our Balanced Hyperbolic Embeddings are better for the task at hand than existing options. In Table 3 (left), we show the out-of-distribution performance. We observe that hierarchical hyperbolic embeddings in general are highly effective for

Table 3: **Comparisons to other hyperbolic approaches.** OOD evaluations when training on CIFAR-100 and scoring with the maximum softmax probability. (Left) Poincaré Embeddings (PE) (Nickel & Kiela, 2017) and Hyperbolic Entailment Cones (HEC) (Ganea et al., 2018) form strong baselines for out-of-distribution, even with low in-distribution performance. This highlights the inherent match of hierarchical hyperbolic embeddings and OOD detection. Our approach remains the strong for both in- and out-of-distribution classification. (Right). Our hyperbolic embeddings are preferred over Clipped Hyperbolic (CH) Guo et al. (2022) classifiers and Poincaré ResNet (PR) (van Spengler et al., 2023).* denotes our re-implementation of the baseline, $\dagger$ denotes results with publicly available pre-trained model.

| Embedding | Dist.↓ | ACC↑ | FPR@95↓ | AUROC↑ | AUPR↑ | Method | FPR@95↓ | AUROC↑ | AUPR↑ |
|---|---|---|---|---|---|---|---|---|---|
| PE | 0.714 | 61.2 | 50.50 | **83.48** | **72.83** | CH$^{\star}$ | 65.38 | 73.38 | 53.93 |
| HEC | 0.172 | 52.1 | 53.18 | 81.92 | 70.63 | PR$^{\dagger}$ | 87.83 | 58.27 | 37.73 |
| **Ours** | **0.026** | **73.4** | **49.46** | 82.43 | 70.41 | **Ours** | **49.46** | **82.43** | **70.41** |

out-of-distribution detection. For FPR@95 for example, we outperform Poincaré Embeddings and Hyperbolic Entailment Cones, but not by a big margin. We also include the in-distribution classification accuracy and the hierarchical distortion rates (Sala et al., 2018) to get the full picture. These values reveal that the baseline embeddings yield a much higher hierarchical distortion than our approach and are actually not well suited for standard classification. In other words, even a suboptimal hierarchical hyperbolic embedding space is a strong out-of-distribution detector. Our Balanced Hyperbolic Embeddings obtain strong out-of-distribution evaluations while maintaining similar in-distribution classification compared to standard softmax cross-entropy training.

**Comparison to Hyperbolic Networks.** The clipped hyperbolic classifiers of Guo et al. (2022) and the Poincaré ResNet of van Spengler et al. (2023) have previously reported out-of-distribution results on OpenOOD. In the fourth experiment, we investigate how well our approach fares compared to the state-of-the-art hyperbolic out-of-distribution approaches. Both baselines rely on the maixmum softmax probability in their work, hence we use the same scoring function for our approach. The results in Table 3 (right) show that our approach is preferred over both alternatives.

**Comparison to SOTA prototype-based methods.** Recent prototype-based approaches like CIDER Ming et al. (2023) and PALM Lu et al. (2024) use class means as prototypes on a hypersphere to learn compact embeddings for OOD. CIDER uses one prototype per class and PALM uses 6 prototypes per class and use MLE to encourage the compactness between samples and the prototypes. Both methods also have an additional contrastive loss to push prototypes far away from each other. In contrast, we predetermine the hy-

Table 4: **Comparison with prototype-based approaches.** KNN scoring function (k=300) $^{\dagger}$ evaluated with publicly available pre-trained models. $^{\star}$ with 128-dim with projection layer and embeddings

| | FPR@95 ↓ | AUROC ↑ | n-AUROC ↑ |
|---|---|---|---|
| CIDER $^{\dagger}$ | 43.24 | 86.18 | 75.43 |
| PALM $^{\dagger}$ | 38.27 | 87.76 | **78.96** |
| Ours $^{\star}$ | **35.83** | **89.45** | 78.50 |

perbolic prototypes based on hierarchy and train with a cross entropy loss based on hyperbolic distances. For fair comparison, we use the same backbone for all methods, ResNet with a 128-dim projection head and use 128-dim hyperbolic prototypes. We show in Table 4 that our method outperforms CIDER and PALM on far-OOD datasets and is on-par with PALM on near-OOD datasets.

**Hierarchical generalization.** To assess how well our method generalizes to unseen data with a closely related hierarchy, we use the five CIFAR-100 OSR splits from OpenOOD Zhang et al. (2023b), defining a hierarchy only for in-distribution classes during training. The evaluation for hierarchical generalization is defined as follows: (1) OOD Detection Granularity: The model's ability to classify the closely related open-set split as OOD is measured on standard OOD benchmark datasets, treating the split as near-OOD. (2) Precision in Hierarchical Relationships: Metrics such as H-Dist Bertinetto et al. (2020) and HSI Dengxiong & Kong (2023) are used to measure how accurately the model identifies the closest related ID class for open-set samples. Detailed metric descriptions are in Appendix C.3.

Table 6: **Hierarchical generalization evaluation on hierarchcical relationships** with H-Dist and HSI for CIFAR-OOD split.

| | H-Dist ↓ | HSI-$b_1$ ↑ | HSI-$b_2$ ↑ |
|---|---|---|---|
| Base | 3.25 | 31.83 | 40.43 |
| Ours | **2.32** | **67.21** | **71.32** |

Table 5: **Hierarchical generalization evaluation on OOD performance.** In-distribution data is from CIFAR-OSR split Zhang et al. (2023b). *All benchmark* compares the performance on far-OOD datasets and AUROC on near-OOD dataset, which includes the OOD split of CIFAR100. *CIFAR-ood-split* reports the full near-OOD performance on the OSR eval split. Hierarchical hyperbolic embeddings perform better on challenging near-OOD splits.

| | FPR@95↓ | | AUROC↑ | | AUPR↑ | | n-AUROC↑ | |
|---|---|---|---|---|---|---|---|---|
| | Base | Ours | Base | Ours | Base | Ours | Base | Ours |
| **All benchmarks** | 55.64 | **44.49** | 78.84 | **84.54** | 57.02 | **65.80** | 79.40 | **81.54** |
| **CIFAR-ood-split** | 59.84 | **54.16** | 77.83 | **82.55** | 75.39 | **78.02** | - | - |

(a) MSP score: baseline (left) vs ours (right).  (b) Energy score: baseline (left) vs ours (right).

Figure 3: **MSP and energy score histograms** for standard deep networks and the same networks with our hyperbolic embeddings. We find that hyperbolic embeddings naturally position out-of-distribution samples farther from in-distribution classes and obtain more easy to discriminate densities, whether only look at the closest in-distribution class (a) or at all classes (b).

In Table 5, we report results averaged over five splits comparing with baseline Euclidean model without any hierarchical information. For far-OOD datasets (MNIST, Textures, SVHN, Places365), we evaluate FPR@95, AUROC, and AUPR. For near-OOD datasets (CIFAR-10, TIN, and CIFAR-OOD split), we report near-AUROC. Specifically, for the CIFAR-OOD split, we report OOD metrics separately to highlight the benefits of incorporating hierarchical information through hyperbolic prototypes. Table 6 evaluates hierarchical precision with H-Dist, which measures the LCA distance between the predicted ID class and ground truth, and HSI, which calculates the inverse of the distance between the LCA and ground truth ancestor ($b_1$), LCA and ground truth class ($b_2$). Higher HSI values indicate better recognition of unknown classes, showcasing the advantages of hyperbolic learning with hierarchical information.

From both tables, we conclude that our method performs well in highly challenging settings (Table 5) and that hierarchical in-distribution training results in better alignment between in- and out-of-distribution classes, even without knowledge of OOD classes (Table 6)

**Analyzing the hyperbolic embeddings,** To better understand the match between our hyperbolic embeddings and out-of-distribution detection, we have performed additional analyses and visualizations. In Figure 3, we show the maximum softmax probability and energy-based histograms for CIFAR-100 (in-distribution) and SVHN (out-of-distribution). We observe that our approach naturally embeds out-of-distribution samples farther from class prototypes. When using the maximum softmax probability as scoring function, nearly all out-of-distribution samples obtain a score below 0.5, making for a stronger separation. The same holds when looking at the entire probability distribution, as done in energy-based scoring. We conclude that our hyperbolic embeddings make it easier to pinpoint out-of-distribution samples, despite being trained on the same in-distribution data, with the same backbone, and the same scoring criteria.

In Figure 4, we show the distribution of ID and OOD samples in the hyperbolic space. We trained a ResNet-34 with 2D hyperbolic embeddings and plot the relative densities of ID and OOD samples. OOD samples mostly have low norm while ID samples are more confident and closer to prototypes near the boundary. This result is in line with other recent findings from hyperbolic learning, indicating that the distance to the edge of the Poincaré ball provides a natural measure of uncertainty.

# 6 RELATED WORK

We briefly introduce recent works that form the motivation for our proposed method and expand on a complete list of related works in Appendix D.

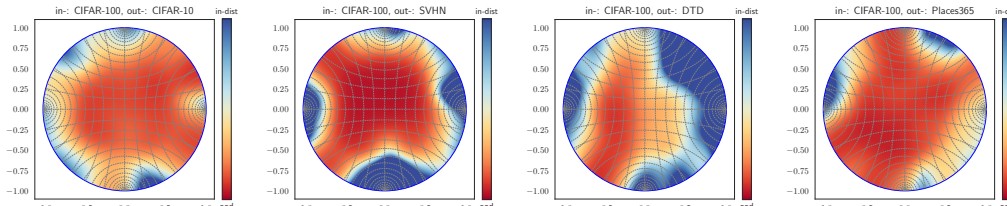

Figure 4: **Visualizing our hyperbolic embeddings in a 2D Poincaré ball.** We plot a relative density heatmap for ID and OOD samples in the Poincaré ball. The **red** areas denote higher concentration of the out-of-distribution samples and **blue** area denotes in-distribution samples.

**Out-of-distribution detection.** Recent methods like CIDER (Ming et al., 2022), PALM (Lu et al., 2024) show that training with hyperspherical prototypes makes the network robust to out-of-distribution samples. where OOD samples lie between ID clusters on the hypersphere. Motivated in a similar way, our method allows OOD samples to additionally lie between ID clusters and origin by choosing hyperbolic geometry. There is some recent exploration into methods that do not just rely on binary out-of-distribution detection. Lee et al. (2018a) introduce hierarchical novelty detection where they aim to find the closest super class for a novel class. This has also been investigated in generalized open-set recognition (Geng et al., 2020; Dengxiong & Kong, 2023), using hierarchies and attributes. In our work, beyond conventional OOD detection, we introduce a fine-grained evaluation approach that leverages hierarchies for improved detection.

**Hyperbolic embeddings of hierarchies.** The foundational work of Nickel and Kiela (Nickel & Kiela, 2017) demonstrated that hyperbolic embeddings outperform Euclidean embeddings for hierarchical data. Extensions include entailment cones for stricter hierarchical relations (Ganea et al., 2018), combinatorial constructions (Sala et al., 2018), and effective applications of the Lorentz model (Nickel & Kiela, 2018; Law et al., 2019).Recent unsupervised metric learning methods (Yan et al., 2021; Kim et al., 2023) were also effective to discover hierarchical information about data. We find that existing embedding algorithms assume balanced hierarchies, resulting in suboptimal embeddings of shallow subhierarchies. We introduce a distortion-based objective with explicit subhierarchy-balancing to avoid this limitation, which directly benefits out-of-distribution detection.

**Hyperbolic learning of visual data.** Hyperbolic learning has shown promise for OOD detection (Guo et al., 2022; van Spengler et al., 2023). Hyperbolic embeddings have been used for generalized open-set recognition (Lee et al., 2018a; Dengxiong & Kong, 2023) and visual anomaly detection (Hong et al., 2023), where OOD samples are naturally positioned near the origin. A similar recent work from Zeng *et al.* (Zeng et al., 2023) show that learning hierarchies through tree distance regularization in euclidean space is beneficial for robustness. We take inspiration such works and strive to balance shallow and wide sub-hierarchies in our hyperbolic embeddings to avoid unwanted biases to outperforms existing hyperbolic out-of-distribution detection approaches. Our approach is general in nature and can be used with any out-of-distribution scoring function.

## 7 CONCLUSIONS

Out-of-distribution detection is a difficult task. This work advocates for hierarchical hyperbolic embeddings to perform such a discrimination. We introduce an algorithm for positioning in-distribution classes as prototypes using their hierarchical relations through a balanced distortion-based objective. In turn, in-distribution learning becomes a hyperbolic sample-to-prototype optimization. Rather than adding yet another score, we show how the well-known existing functions effortlessly generalize to operate with hyperbolic prototypes. Experiments across a wide range of datasets and scoring functions highlights the strong potential of hyperbolic embeddings for out-of-distribution detection. We furthermore show that our approach leads to hierarchical out-of-distribution generalization without any knowledge about out-of-distribution classes. We conclude that Balanced Hyperbolic Learning is a powerful, general-purpose approach to enrich your out-of-distribution detection. **Limitations.** We assume that a correct and known hierarchy is available. While it is possible to use LLM-generated hierarchies Liu et al. (2024), verifying the correctness and usability is an exciting direction for future work.

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

## A   MOTIVATION FOR BALANCED HYPERBOLIC EMBEDDINGS

**On bias towards deeper and wider subtrees.** To better understand bias in existing methods towards imbalances in hierarchies, we construct an imbalanced hierarchy over CIFAR-100 for 2,3 and 4 levels of granularity. This hierarchy deliberately incorporates subtrees of varying depths ( *i.e.* levels of hierarchy) and widths (*i.e.* number of nodes), allowing us to systematically analyze how different approaches learn embeddings across uneven hierarchies. Specifically, we compare the learned hierarchies from three methods: Poincaré embeddings (PE) Nickel & Kiela (2017), Hyperbolic entailment cones (HEC) Ganea et al. (2018), and our proposed balanced hyperbolic embeddings. To analyze these methods, we plot the pairwise distances between nodes in the hierarchy, as shown in Figure 5. These pairwise distance plots help visualize the structural relationships within the learned embeddings, including the granularity and differentiation between hierarchical levels.

The visualizations reveal that existing methods such as PE and HEC exhibit a tendency to over-prioritize narrower subtrees (those with fewer nodes) compared to wider subtrees, especially as granularity increases. Moreover, these methods display limited differentiation between deeper levels of hierarchy, as evidenced by lower color gradient between leaf nodes (diagonal) and their corresponding parent nodes in the pairwise distance plot. Our proposed approach, on the other hand, demonstrates a more balanced representation, effectively addressing these biases, providing a more accurate representation of the hierarchical structure.

**Motivation for losses.** The distortion loss ensures that all in-distribution classes are distributed in a uniform hierarchical manner. The norm loss ensures that all nodes at the same hierarchical level are equally far away from the origin. This is highly preferred for OOD, especially when dealing with imbalanced trees, as OOD samples tend to be embedded closer to the origin. With our norm loss, we avoid a bias of OOD samples to shallow subtrees, leading to better ID/OOD discrimination. We visualize the variance of norms across all hierachical levels for a toy tree example to explain our point. For a balanced tree with 3 levels and 5 nodes per level, we remove a percentage of nodes randomly to introduce imbalance. In Figure 6, we plot the variance of norms as a function of the percentage of nodes removed comparing our approach with distortion loss alone to the combination of distortion and norm loss. The results clearly demonstrate that without the norm loss, the variance of the norms increases significantly in imbalanced hierarchies, thereby underscoring the role of norm loss in achieving balanced hierarchical representations.

## B   EVALUATING THE QUALITY OF BALANCED HYPERBOLIC EMBEDDINGS

**Visualizing learnt hierarchies.** Figure 7 depicts the hierarchies learnt for CIFAR-100 and ImageNet-100 datasets and average norms of each level of the hierarchy. These visualizations consist of three components for each dataset: the structure of the learned hierarchical tree, the pairwise hyperbolic distance matrix between the graph nodes, and the average norm of samples at each level of the hierarchy. Overall, the visualizations demonstrate the effectiveness of our approach in learning fair approximations of hierarchies in hyperbolic space.

**Ablation of embedding dimensions.** The embedding dimensionality is a hyperparameter that can be freely set. In Table 7, we show how well the graph distances are preserved using the distortion and MAP metrics of Sala et al. Sala et al. (2018). We find that our approach is highly stable, with a small preference for 64 dimensions.

Table 7: Embedding quality as a function of embedding dimensions on CIFAR-100.

| Emb. dim. | 8 | 16 | 32 | 64 | 128 | 256 |
|---|---|---|---|---|---|---|
| MAP $\uparrow$ | 0.84 | 0.84 | 0.86 | **0.88** | 0.86 | 0.86 |
| Distortion $\downarrow$ | 0.054 | 0.029 | 0.028 | **0.026** | 0.026 | 0.026 |

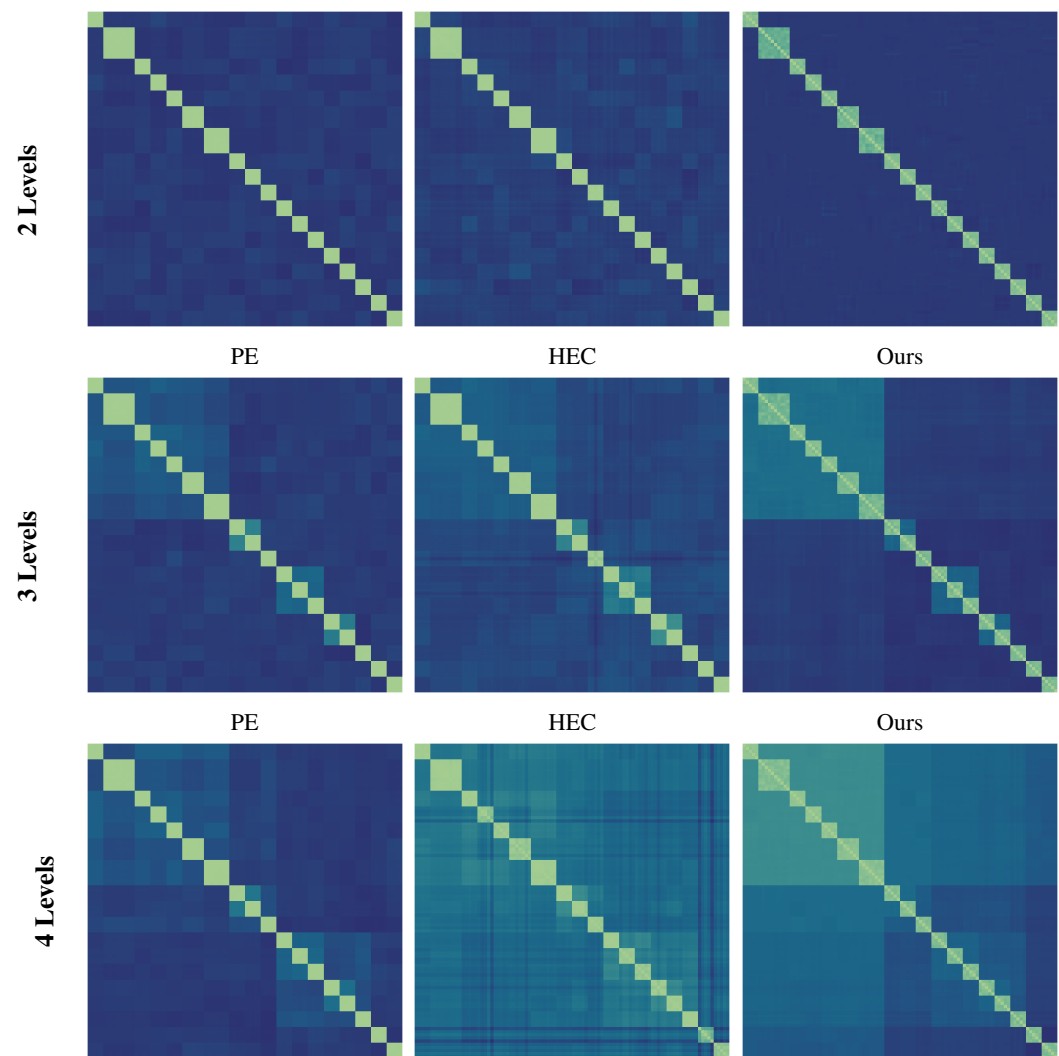

Figure 5: **Stability in the face of bias.** Pairwise distance plots across different levels of granularity for an imbalanced CIFAR-100 graph. Lighter distances are closer in the embedding space compared to darker distances. Showing (left) Poincaré embeddings Nickel & Kiela (2017), (middle) Hyperbolic entailment cones Ganea et al. (2018) and our (right) balanced hyperbolic embeddings. Our method is better at reconstructing the hierarchy, especially for imbalanced deeper hierarchies.

## C   ADDITIONAL EXPERIMENTAL RESULTS

### C.1   EXPERIMENTAL SETUP FOR EUCLIDEAN BASELINE

For CIFAR-100 and ImageNet-100, we train a ResNet-34 for 200 epochs trained with cross entropy loss. The batch size is 128 for CIFAR and 256 for ImageNet. We use SGD with 0.9 momentum and a learning rate of 0.1 with cosine annealing scheduler (Loshchilov & Hutter, 2016), with a weight decay of 0.0005. We perform 3 independent training runs for each method and report the average performance.

### C.2   EXPERIMENTS (CONTD.)

**Norms in ID vs OOD embeddings.** We plot the distribution of hyperbolic norms, (Eq. 3) $d_{\mathbb{B}}(\mathbf{x}, 0)$, for in-distribution (ID) vs out-of-distribution (OOD) samples to visualize the separation between the embeddings based on the norm of the samples (Figure 8). As expected, we observe that the norms

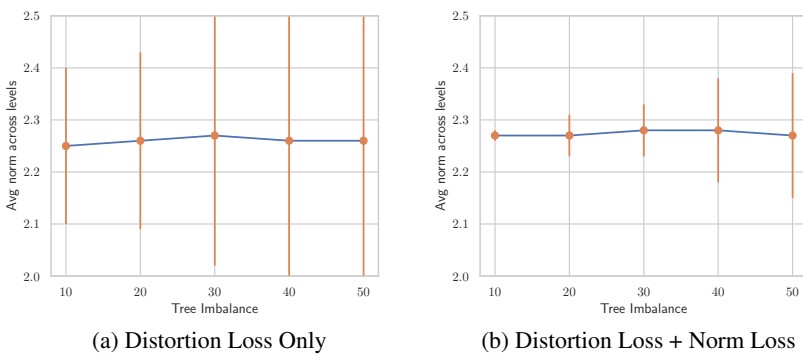

(a) Distortion Loss Only  (b) Distortion Loss + Norm Loss

Figure 6: Variance in norms with and without balancing for increasing tree imbalance. Adding norm loss (b) leads to consistent norms across all levels, compared to distortion loss alone (a).

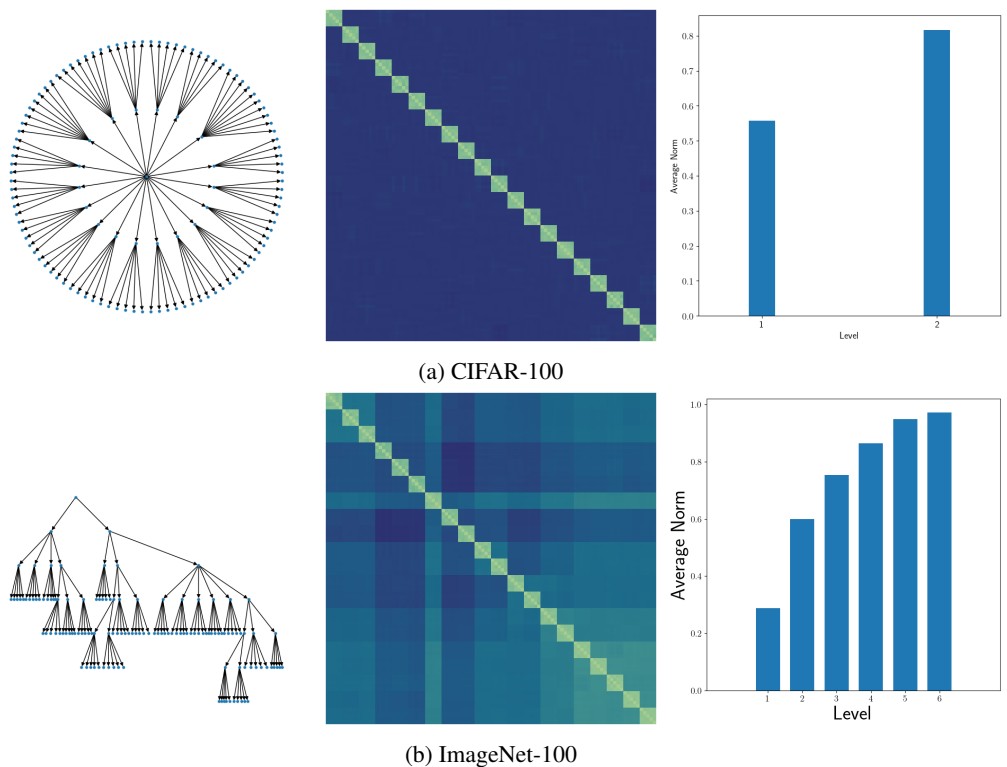

(a) CIFAR-100

(b) ImageNet-100

Figure 7: Hierarchies learnt in CIFAR-100 (7a) and ImageNet-100 (7b). (Left) Tree of the hierarchy, (Middle) Plot of pairwise hyperbolic distances between each nodes of the graph to illustrate the learned hierarchy. Lighter distances are closer in the embedding space compared to darker distances. (Right) Average norm of samples at each level of the hierarchy.

of ID samples are generally high, indicating that these points closer to the boundary of the Poincaré ball. In contrast, most OOD samples exhibit lower norm, positioning them closer to the origin.

**Additional backbones.** We show results on other common backbones in OOD literature, WideResNet and DenseNet-BC in Figure 9 for MSP and KNN. We find that for all backbones, our balanced hyperbolic learning outperforms the Euclidean baseline across scoring functions.

**Expanded results Imagenet-100.** We expand on Table 2 for additional scoring functions below in Table 8. This confirms the primary observations, further highlighting the versatility of hyperbolic embeddings under various scoring settings.

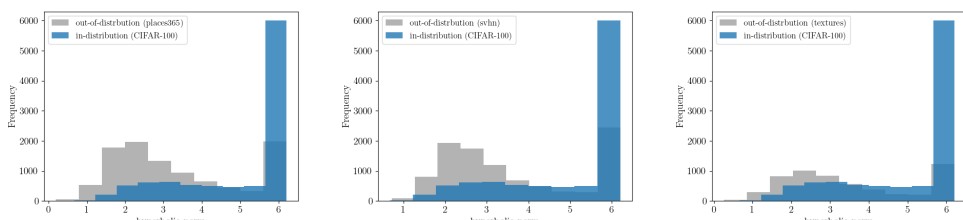

Figure 8: Hyperbolic norms across in-distribution (CIFAR-100) and various out-of-distribution (OOD) datasets. Most OOD samples can be easily identified based on their distance to the origin.

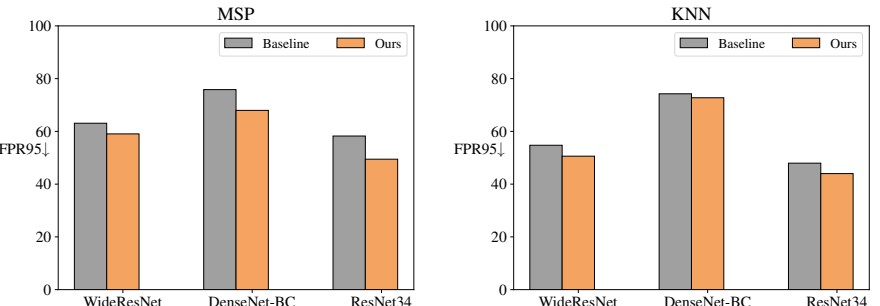

Figure 9: FPR95 for the Euclidean baseline and ours with different backbones on CIFAR-100, with MSP (left) and KNN (right) as scoring functions.

**Dataset wise results OOD.** We expand on the dataset-specific results corresponding to our main table (Table 1) for out-of-distribution (OOD) evaluation when the model is trained on is CIFAR-100 as in-distrubution data (see Table 9). To outline, we employed a ResNet-34 trained on CIFAR-100 for 200 epochs. In the baseline approach, the model is trained with a cross-entropy loss. In our proposed method, we project the features of the last layer into a Poincaré ball and compute distances to prototypes derived from Balanced Hyperbolic Embedding training, as outlined in Section 3.2 of the main text, and trained using cross-entropy loss. The far-OOD evaluation datasets are MNIST, SVHN, Textures and Places 365.

## C.3 ABLATIONS

**AUPR and AUROC.** Continuation of Figure 2, where we report the ablations of euclidean and hyperbolic approaches for OOD on FPR@95 and AUROC, in Figure 10 we report the AUPR and near-AUROC. These metrics follow the same trends observed in earlier reported metrics, demonstrating that balanced hierarchical embeddings consistently lead to the best OOD performance.

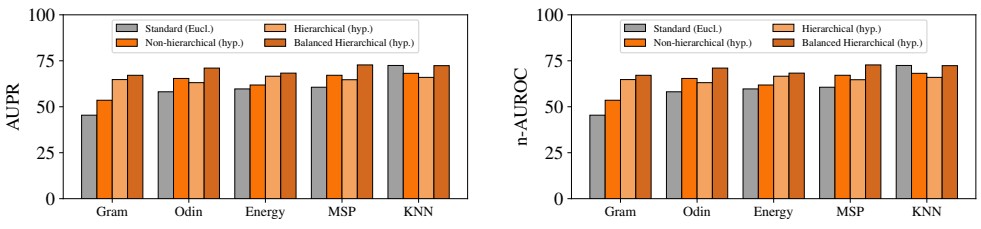

Figure 10: **Out-of-distribution ablation study** for AUPR ↑(left) and AUROC ↑ (right).

**Curvature of Hyperbolic Space.** To investigate the impact of curcature on OOD detection performance, conduct an ablation study by varying the curvature parameter $c$ for network trained with

Table 8: **Balanced Hyperbolic Learning 8 functions** evaluated on OpenOOD with ImageNet100, extension of Table 2

| | FPR@95 ↓ | | AUROC ↑ | | AUPR ↑ | | n-AUROC ↑ | |
|---|---|---|---|---|---|---|---|---|
| | Base | Ours | Base | Ours | Base | Ours | Base | Ours |
| **DICE** Sun & Li (2022) | 38.51 | **37.31** | 88.10 | **89.10** | 76.54 | **79.33** | 80.12 | **80.19** |
| **RankFeat** Song et al. (2022) | 98.72 | **74.82** | 36.12 | **73.13** | 45.89 | **58.17** | 50.71 | **57.84** |
| **ASH** Djurisic et al. (2022) | 32.44 | **27.84** | 90.41 | **91.02** | 75.64 | **76.87** | 79.84 | 78.12 |
| **SHE** Zhang et al. (2022b) | 46.18 | **37.54** | 86.80 | **89.31** | 70.23 | **75.64** | 74.56 | **77.26** |
| **GEN** Liu et al. (2023) | **37.10** | 37.25 | 89.92 | **89.96** | 76.21 | **77.15** | **81.04** | 80.24 |
| **NNGuide** Park et al. (2023) | 31.84 | **27.21** | 90.12 | **91.24** | 74.56 | **76.38** | 82.34 | **83.11** |
| **SCALE** Xu et al. (2023) | 26.31 | **25.69** | **88.14** | 86.21 | **75.89** | 73.44 | **80.81** | 79.83 |

Table 9: **Dataset-wise results on far-OOD datasets.** The model is trained on CIFAR-100 and evaluated on four far-OOD datasets: MNIST, SVHN, Textures and Places 365. 9a shows FPR ↓ performance and 9b shows AUROC ↑ performance across the datasets.

(a) FPR95 ↓

| | MNIST | | SVHN | | Textures | | Places 365 | | Average | |
|---|---|---|---|---|---|---|---|---|---|---|
| | Base | Ours | Base | Ours | Base | Ours | Base | Ours | Base | Ours |
| **MSP** | 64.70±1.50 | **56.91±1.59** | 46.0 ± 0.08 | **27.65± 0.20** | 61.25 ± 0.06 | **53.69 ± 0.25** | 60.99 ± 0.87 | **59.59 ± 0.22** | 58.24 ± 1.08 | **49.46 ± 0.23** |
| **TempScale** | 64.38 ± 1.67 | **56.52 ± 1.64** | 44.02 ± 0.05 | **25.43 ± 0.21** | 60.58 ± 0.08 | **53.11 ± 0.24** | 61.21 ± 0.92 | **59.37 ± 0.17** | 57.55 ± 1.21 | **48.61 ± 0.24** |
| **Odin** | 63.37 ± 1.72 | **55.03 ± 1.93** | 56.52 ± 0.14 | **28.64 ± 1.16** | 60.17 ± 0.85 | **52.37 ± 0.19** | 63.76 ± 0.92 | **61.78 ± 0.59** | 60.96 ± 1.36 | **49.45 ± 0.34** |
| **Gram** | 85.82 ± 2.36 | **55.30 ± 4.19** | 62.18 ± 4.08 | **22.70 ± 1.18** | 89.61 ± 3.46 | **68.48 ± 3.48** | 95.73 ± 1.63 | **84.61 ± 0.48** | 83.33 ± 0.28 | **57.77 ± 0.31** |
| **Energy** | **66.95 ± 1.79** | 70.21 ± 2.43 | 40.66 ± 0.03 | **29.93 ± 3.36** | 60.83 ± 0.08 | **52.49 ± 5.27** | 65.43 ± 1.25 | **68.99 ± 3.47** | 58.47 ± 1.34 | **55.41 ± 1.30** |
| **KNN** | 52.39 ± 2.20 | **51.06 ± 0.60** | 30.80 ± 2.00 | **20.34 ± 2.55** | 53.29 ± 1.50 | **46.67 ± 3.58** | 60.21 ± 4.13 | **57.93 ± 1.05** | 49.17 ± 0.68 | **44.00 ± 0.10** |
| **DICE** | 67.36 ± 5.01 | **67.51 ± 3.06** | 40.91 ± 8.97 | **28.34 ± 5.58** | 63.88 ± 2.83 | **56.94 ± 0.98** | 65.34 ± 2.50 | **65.89 ± 0.67** | 59.37 ± 0.57 | **54.67 ± 0.24** |
| **Rank Feat** | 73.62 ± 1.01 | **53.26 ± 1.20** | 63.64 ± 5.86 | **37.36 ± 1.54** | 68.94 ± 5.02 | **37.12 ± 4.09** | 85.91 ± 2.30 | **71.89 ± 0.71** | 73.03 ± 1.85 | **49.91 ± 0.33** |
| **ASH** | 79.13 ± 0.93 | **61.82 ± 0.17** | 49.66 ± 2.08 | **34.45 ± 2.25** | 64.57 ± 3.34 | **58.45 ± 1.29** | 76.56 ± 0.19 | **66.42 ± 3.95** | 67.48 ± 0.73 | **55.29 ± 0.04** |
| **SHE** | 87.45 ± 2.89 | **64.67 ± 0.64** | 58.07 ± 2.03 | **34.34 ± 3.21** | 80.38 ± 0.69 | **48.47 ± 3.66** | 82.38 ± 0.37 | **67.65 ± 0.55** | 77.07 ± 0.43 | **53.78 ± 0.25** |
| **GEN** | 60.89 ± 1.81 | **56.78 ± 0.15** | 40.18 ± 0.04 | **24.96 ± 1.37** | 58.72 ± 0.36 | **53.53 ± 0.28** | 58.86 ± 0.74 | **59.53 ± 0.30** | 54.66 ± 1.37 | **48.70 ± 0.35** |
| **NNGuide** | 76.71 ± 0.83 | **72.45 ± 1.30** | 52.93 ± 0.14 | **26.94 ± 0.03** | 68.09 ± 0.39 | **59.02 ± 1.56** | 64.02 ± 2.34 | 73.34 ± 1.79 | 65.44 ± 0.58 | **57.94 ± 1.02** |
| **SCALE** | 66.60 ± 1.87 | **60.53 ± 0.23** | 40.89 ± 0.07 | **32.44 ± 0.34** | 56.59 ± 1.40 | **55.99 ± 1.20** | 66.53 ± 0.67 | **64.50 ± 0.03** | 57.65 ± 1.07 | **53.36 ± 0.15** |

(b) AUROC ↑

| | MNIST | | SVHN | | Textures | | Places 365 | | Average | |
|---|---|---|---|---|---|---|---|---|---|---|
| | Base | Ours | Base | Ours | Base | Ours | Base | Ours | Base | Ours |
| **MSP** | 69.9 ± 0.67 | **75.42 ± 0.21** | 84.79 ± 0.01 | **93.02 ± 0.23** | 76.60 ± 0.19 | **81.78 ± 0.04** | 76.91 ± 0.09 | **78.49 ± 0.18** | 77.05 ± 0.48 | **82.43 ± 0.26** |
| **TempScale** | 70.98 ± 0.82 | **76.77 ± 1.22** | 86.31 ± 0.02 | **94.31 ± 0.17** | 77.86 ± 0.21 | **82.24 ± 0.02** | 77.58 ± 0.08 | **78.80 ± 0.13** | 78.18 ± 0.59 | **83.02 ± 0.16** |
| **Odin** | 72.84 ± 1.05 | **78.21 ± 0.29** | 77.77 ± 0.29 | **92.16 ± 0.43** | 78.77 ± 0.19 | **82.97 ± 0.26** | 77.15 ± 0.15 | **78.52 ± 0.28** | 76.63 ± 0.83 | **82.97 ± 1.24** |
| **Gram** | 54.29 ± 1.33 | **70.68 ± 1.56** | 81.76 ± 0.58 | **95.19 ± 0.53** | 69.95 ± 3.83 | **83.05 ± 0.79** | 43.22 ± 1.76 | **58.42 ± 0.23** | 62.31 ± 0.36 | **76.84 ± 1.15** |
| **Energy** | 71.01 ± 1.32 | **77.07 ± 0.10** | 87.51 ± 0.07 | **88.58 ± 0.38** | 78.79 ± 0.14 | **82.21 ± 0.97** | 76.21 ± 0.23 | **78.85 ± 0.08** | 78.38 ± 0.99 | **81.74 ± 0.50** |
| **KNN** | 76.66 ± 4.78 | **80.26 ± 1.93** | 91.85 ± 0.27 | **95.91 ± 0.08** | 83.33 ± 2.33 | **86.00 ± 0.36** | 78.53 ± 0.24 | **79.79 ± 0.24** | 82.59 ± 0.28 | **85.51 ± 0.17** |
| **DICE** | **72.44 ± 0.25** | 72.17 ± 0.34 | 87.27 ± 1.81 | **93.06 ± 0.19** | 77.27 ± 1.66 | **81.27 ± 2.40** | 74.88 ± 1.21 | **77.33 ± 1.68** | 77.96 ± 1.12 | **80.96 ± 1.73** |
| **Rank Feat** | 72.75 ± 0.11 | **80.15 ± 0.68** | 74.63 ± 4.55 | **85.35 ± 0.64** | 74.07 ± 5.37 | **90.58 ± 0.64** | 54.48 ± 10.24 | **68.93 ± 1.17** | 68.98 ± 1.13 | **81.25 ± 0.35** |
| **ASH** | 68.18 ± 0.83 | **73.46 ± 0.17** | 86.47 ± 0.16 | **85.89 ± 0.02** | **80.08 ± 0.19** | 75.76 ± 2.34 | **72.77 ± 0.02** | 72.21 ± 1.22 | **76.88 ± 0.55** | 76.83 ± 0.51 |
| **SHE** | 55.74 ± 2.87 | **76.58 ± 1.61** | 80.85 ± 0.92 | **90.32 ± 0.58** | 67.83 ± 0.21 | **85.13 ± 0.65** | 63.95 ± 1.27 | **78.53 ± 0.33** | 67.09 ± 2.26 | **82.02 ± 0.60** |
| **GEN** | 72.78 ± 0.68 | **76.73 ± 0.01** | 86.61 ± 0.01 | **94.25 ± 1.08** | 78.62 ± 0.22 | **82.18 ± 0.11** | 78.82 ± 0.08 | 78.65 ± 0.25 | 79.21 ± 0.49 | **82.95 ± 0.40** |
| **NNGuide** | 63.97 ± 1.03 | **76.08 ± 0.20** | 85.76 ± 0.70 | **88.45 ± 0.14** | 77.57 ± 0.70 | **81.98 ± 1.42** | 78.19 ± 1.05 | **78.39 ± 0.58** | 76.37 ± 0.80 | **81.23 ± 0.99** |
| **SCALE** | 71.86 ± 1.21 | **76.66 ± 1.67** | 88.37 ± 0.02 | 86.89 ± 0.33 | 81.82 ± 0.03 | 78.38 ± 1.34 | 76.65 ± 0.15 | 74.85 ± 0.43 | **79.67 ± 0.78** | 79.20 ± 0.79 |

(c) AUPR ↑

| | MNIST | | SVHN | | Textures | | Places 365 | | Average | |
|---|---|---|---|---|---|---|---|---|---|---|
| | Base | Ours | Base | Ours | Base | Ours | Base | Ours | Base | Ours |
| **MSP** | 40.95 ± 1.43 | **48.15 ± 5.58** | 74.69 ± 0.01 | **87.07 ± 0.72** | 85.31 ± 0.03 | **88.56 ± 0.03** | 56.54 ± 0.51 | **57.84 ± 0.11** | 64.37 ± 1.24 | **70.41 ± 0.39** |
| **TempScale** | 41.43 ± 1.38 | **48.66 ± 0.77** | 76.52 ± 0.00 | **88.85 ± 0.56** | 86.01 ± 0.04 | **88.84 ± 0.03** | 56.79 ± 0.80 | **58.17 ± 0.10** | 65.18 ± 1.39 | **71.13 ± 0.40** |
| **Odin** | 42.84 ± 1.92 | **49.78 ± 1.19** | 65.49 ± 0.27 | **85.67 ± 2.31** | 86.44 ± 0.06 | **89.25 ± 0.10** | 55.18 ± 1.49 | **56.34 ± 0.21** | 62.48 ± 1.56 | **70.26 ± 0.91** |
| **Gram** | 18.02 ± 5.62 | **50.07 ± 4.59** | 63.57 ± 1.94 | **89.25 ± 3.07** | 74.08 ± 2.82 | **86.81 ± 0.33** | 18.67 ± 0.94 | **32.45 ± 1.23** | 43.58 ± 2.62 | **64.64 ± 5.13** |
| **Energy** | **38.92 ± 2.61** | 29.80 ± 19.85 | 78.24 ± 0.11 | **79.45 ± 11.83** | 86.37 ± 0.02 | **85.82 ± 1.53** | 53.65 ± 2.42 | **48.84 ± 1.79** | 64.30 ± 1.14 | **61.83 ± 5.06** |
| **KNN** | **55.69 ± 1.17** | 54.07 ± 1.09 | 85.45 ± 0.77 | **91.23 ± 0.36** | 89.43 ± 1.10 | **91.04 ± 0.20** | 58.96 ± 2.49 | 57.87 ± 1.65 | 71.02 ± 1.55 | **73.71 ± 1.44** |
| **DICE** | 30.43 ± 3.31 | **38.46 ± 4.77** | 73.92 ± 0.32 | **86.06 ± 2.24** | 83.59 ± 0.07 | **87.73 ± 1.45** | 49.80 ± 3.44 | **53.16 ± 3.20** | 59.43 ± 2.36 | **66.35 ± 3.74** |
| **Rank Feat** | 35.10 ± 5.63 | **54.97 ± 6.26** | 60.12 ± 2.24 | **79.30 ± 8.69** | 82.53 ± 20.71 | **93.97 ± 0.03** | 29.80 ± 5.08 | **45.81 ± 7.52** | 51.89 ± 10.90 | **68.51 ± 2.98** |
| **ASH** | 26.09 ± 6.08 | **45.79 ± 4.61** | 72.71 ± 0.61 | **80.57 ± 0.08** | 86.46 ± 0.02 | **84.95 ± 0.27** | 44.44 ± 0.27 | **50.48 ± 5.28** | 57.43 ± 2.46 | **64.89 ± 0.86** |
| **SHE** | 18.82 ± 5.19 | **37.96 ± 6.93** | 66.28 ± 2.02 | **88.18 ± 2.45** | 77.30 ± 0.15 | **87.49 ± 0.46** | 35.92 ± 5.17 | **55.20 ± 1.35** | 49.58 ± 4.06 | **67.21 ± 5.66** |
| **GEN** | 45.08 ± 9.90 | **48.41 ± 4.96** | 78.32 ± 0.00 | **88.67 ± 2.99** | 86.67 ± 0.07 | **88.79 ± 0.06** | 58.94 ± 1.03 | 58.03 ± 0.22 | 67.25 ± 8.33 | **70.98 ± 0.43** |
| **NNGuide** | 30.25 ± 11.94 | **32.74 ± 5.59** | 70.74 ± 8.50 | **83.17 ± 3.50** | 84.47 ± 0.08 | **87.13 ± 1.25** | 56.77 ± 2.19 | 49.49 ± 3.17 | 60.56 ± 3.60 | **63.13 ± 4.77** |
| **SCALE** | 39.47 ± 7.32 | **46.26 ± 2.56** | 78.48 ± 0.06 | **81.35 ± 1.26** | 88.28 ± 0.03 | **86.69 ± 0.76** | 53.13 ± 1.66 | **52.54 ± 0.15** | 67.88 ± 4.11 | **66.71 ± 1.19** |

hyperbolic embeddings on CIFAR-100. We evaluate the OOD performance across the benchmark datasets: CIFAR-10, TinyImageNet as near-OOD and MNIST, SVHN, Places-365 and Textures as far-OOD datasets. The results are in Table 10.

From the table we observe that smaller curvatures (*e.g.* $c = 0.5$) achieve relatively good ID performance but do not excel in OOD detection. Larger curvatures lead to noticeable degradation in both ID and OOD performance. Our method, with $c = 1$ achieves the best results across all metrics.

Table 10: **Ablation of hyperbolic curvature** for CIFAR-100, with reported OOD performance using MSP scoring. We show that c=1 is beneficial for this dataset and generalize it to other datasets.

| curvature | ID acc | FPR95 | AUROC | AUPR | n-AUROC |
|---|---|---|---|---|---|
| 0.5 | 72.36 | 73.40 | 76.91 | 55.7 | 75.24 |
| 0.75 | 71.91 | 74.99 | 76.99 | 54.47 | 74.41 |
| 1.5 | 69.81 | 82.31 | 71.00 | 48.51 | 71.67 |
| 2.0 | 68.89 | 85.29 | 69.79 | 46.49 | 70.62 |
| Ours (c=1) | **73.20** | **49.46** | **82.43** | **70.41** | **78.01** |

### C.4 ABOUT HIERARCHICAL GENERALIZATION

The setting for hierarchical generalization aims to evaluate how well our proposed model can handle OOD samples that belong to a closely related hierarchy. To this end, we adopt the CIFAR-100 OSR 50/50 split setting from OpenOOD [1] Zhang et al. (2023b) and only use hierarchy information for the training data. For the evaluation of hierarchical metrics in Table 6, we use the whole hierarchy to measure the Lowest Common Ancestor (LCA) distances during evaluation. Below we give a detailed description of the hierarchical metrics used.

**Hierarchical Distance (H-Dist).** The H-Dist metric, as defined by Bertinetto et al. (2020), calculates the mean height of the LCA between the ground truth and the predicted class when the input is misclassified. Here, we adapt this metric to consider H-dist as the mean height between LCA and the predicted ID class for an OOD sample.

**Hierarchical Similarity Index (HSI).** We adapt the HSI metric from Dengxiong & Kong (2023) originally proposed for generalized open-set recognition(G-OSR), to fit our hierarchical OOD detection setup. While G-OSR focuses on identifying the closest ancestor for unseen samples from ancestor nodes, our approach instead evaluates how closely the predicted ID class aligns with the true hierarchy of the OOD samples. The metrics are summarized as follows:

$$\text{HSI-}b_1 = \frac{1}{m} \sum_{l=1}^{m} \frac{1}{d(y_{gt1}^l, y_{\text{LCA1}}^l)} \tag{13}$$

$$\text{HSI-}b_2 = \frac{1}{m} \sum_{l=1}^{m} \frac{1}{ln(d(y_{gt2}^l, y_{\text{LCA2}}^l) + 1)e} \tag{14}$$

The hierarchical similarity index is defined by the Lowest Common Ancestor (LCA) distance between ground truth and the direct ances- tor of the predicted class. HSI-$b_1$ is the inverse of distance between direct ground truth ancestor and the lowest common ancestor and HSI-$b_2$ is the inverse of the distance between ground truth class and lowest common ancestor. A lower distance represents better result.

## D RELATED WORK

**Out-of-distribution Detection.** Conventional out-of-distribution detection is viewed as a binary task; a sample is either from the same distribution as the one used during training or not. It was addressed early on by Hendrycks & Gimpel (2016) which proposed a score based on softmax output to detect such samples. Since then, numerous methods have been proposed to address this problem, aiming to utilize confidence and score-based (Hendrycks & Gimpel, 2016; Lee et al., 2018b; Liang et al., 2018; Liu et al., 2020b), distance-based (Lee et al., 2018b; Sehwag et al., 2021; Tao et al., 2022; Sun et al., 2022) or generative-based (Ryu et al., 2018; Kong & Ramanan, 2021) methods to reliably classify whether a sample is out-of-distribution or not. Training-time methods additionally train with outlier data or have additional training strategies to make the network robust to outliers.

---

[1]https://github.com/Jingkang50/OpenOOD/tree/main/configs/datasets/osr_cifar50

Methods that use non-overlapping outlier-data (Liu et al., 2020b; Yu & Aizawa, 2019; Yang et al., 2021; Zhang et al., 2023a) and that generate outlier-data (Kong & Ramanan, 2021) fine-tune the model on the outlier data which makes the model robust to other unseen outliers. Training-time methods like LogitNorm (Wei et al., 2022) and Decoupled Max Logit (Zhang & Xiang, 2023) reformulate logits and derive new training losses. Similarly G-ODIN(Hsu et al., 2020) decompose confidence scoring and modify input pre-processing. Sehwag et al. (2021) and Winkens et al. (2020) train with contrastive losses for better out-of-distribution generalization.

**Hyperbolic learning of visual data.** Hyperbolic learning is quickly gaining traction in deep learning, with applications and new possibilities on various problems, as highlighted in recent surveys (Mettes et al., 2023; Peng et al., 2021). Hyperbolic learning has shown to be beneficial for few-shot learning (Cui et al., 2023; Gao et al., 2021; Khrulkov et al., 2020; Ma et al., 2022; Zhang et al., 2022a), hierarchical recognition (Ghadimi Atigh et al., 2021; Dhall et al., 2020; Liu et al., 2020a; Yu et al., 2022), retrieval (Desai et al., 2023; Ermolov et al., 2022; Long et al., 2020), dealing with uncertainty (Ghadimi Atigh et al., 2022; Franco et al., 2023; Surís et al., 2021), generative learning on scarce data (Bose et al., 2020; Hsu et al., 2021; Li et al., 2022; Mathieu et al., 2019; Nagano et al., 2019), and more.

