# OpenReview forum: "Balanced Hyperbolic Embeddings Are Natural Out-of-Distribution Detectors"
_ICLR.cc/2025/Conference — Submitted to ICLR 2025_

### Official Review · Reviewer_ZNiu · 2024-10-31

**Soundness:** 2
**Presentation:** 2
**Contribution:** 2
**Rating:** 3
**Confidence:** 5

**Summary:**

The paper introduces balanced hyperbolic learning for improved OOD detection by leveraging hyperbolic embeddings to represent class hierarchies. The authors propose a method to embed classes as hyperbolic prototypes that capture hierarchical relationships and seek to balance shallow and wider subtrees in the learned embedding with a distortion loss and norm minimizing term. This method can be easily integrated into existing OOD detection pipelines with minimal modification and works effectively in a wide range of datasets.

---

**Post Rebuttal** I want to first thank the authors for engaging in the discussion period. After the rebuttal discussion, I am more certain that the norm loss does not directly relate to the InD/OOD position. Thus why norm loss is helpful for OOD detection tasks is still not unclear. Even if I assume empirically *"without a norm loss, shallow subtrees also gravitate towards the origin, leading to unwanted biases."* mathematically norm loss does not directly control the distance of nodes to the origin (which is the major motivation in the introduction), putting one of the main contributions of this paper in doubt. Besides, in the current submission, the empirical evidence of *"without a norm loss, shallow subtrees also gravitate towards the origin, leading to unwanted biases."* seems missing. Also, the current paper does not provide a correct definition of what they mean by "levels". This will cause complete confusion for the methodology and many analyses provided in the paper, unfortunately. In summary, I do not think this submission is ready for publication. I will keep my original score and raise the confidence score to 5.

**Strengths:**

The writing of the methodology section is clear. The method is simple and flexible, which means it can be widely applicable to many OOD scores and settings.

**Weaknesses:**

- Novelty is somewhat limited: “the hyperbolic classifier provides strongly uniform distributions for samples near the origin and strongly peaked distributions for samples near the boundary,” this motivation of this paper can date back to at least [1] Fig. 1. [1] uses distance to hyperbolic disk origin as an uncertainty metric in Sec.5.1, which can be also used for OOD detection, without leveraging any training labels. Please cite [1] accordingly in the introduction section and treat it as an important baseline for comparison. Also, the idea of leveraging hierarchical relationships for OOD detection is old, such as [2]. Therefore, Sec 1 and 2 are mostly from previous literature.
- Writing for the experimental section can be improved. In Section 5, because you have multiple paragraphs, it might be better to include a few sentences for summary about what experiments/settings you are going to cover at the beginning of Section 5. Some paragraph names probably need to be revised: for example, “comparison to hyperbolic embeddings” sounds similar to “comparison to other hyperbolic methods”, which is very confusing while reading.
- Soundness of core methodology: the motivation of using the exact formulation of distortion loss and particularly norm loss is questionable. It requires either more explanation or more experimental support. Also, there are many details below that need to be clarified in the experimental section for OOD detection.
- Lacking comparison with some important baselines for the choice of distortion loss functions and hyperbolic neural networks. See Questions section for more details.

[1] Khrulkov, Valentin, et al. "Hyperbolic image embeddings." Proceedings of the IEEE/CVF conference on computer vision and pattern recognition. 2020.

[2] Linderman, Randolph, et al. "Fine-grain inference on out-of-distribution data with hierarchical classification." Conference on Lifelong Learning Agents. PMLR, 2023.

**Questions:**

- Line 70: “distance to the class prototype” how is prototype defined on Poincare disk? Also, how are Euclidean logits, based on dot products with Euclidean classifiers, and Poincare logits comparable? How to compare logits derived from different hyperbolic methods, especially Eq. 11 and those in Hyperbolic Neural Networks such as [1]? This is important because you rely on logit values heavily for many OOD detection scores as you described in Sec. 3.3. Why do you specifically use Eq. 11 as your classification loss? It would be interesting to explore the impact of loss function on hyperbolic OOD detection.
- Line 141-144: How does the quality of the hierarchy affect your method? How do you verify if LLM-generated hierarchies are reliable? Why can you directly transfer WordNet hierarchies to image datasets, because some semantic relationships are not reflected in images. For example, “large/small” in CIFAR100 hierarchy’s coarse label.
- Line 146: what does “equivalent edge distances” mean?
- Line 157-159: Distortion Loss’s motivation is the same as [2] by simply replacing Euclidean feature distance in [2] with Poincare distance, please include the comparison of Eq.7 and CPCC loss. Besides, why do you divide d_G in the loss instead of directly minimizing the l2 distance between two distance metrics?
- Norm loss: What’s the motivation for “nodes on the same level in the hierarchy should have the same norm, ensuring a uniform distribution across levels”? How does this affect OOD detection? I would recommend you to visualize the learned embeddings (like in [3]) of different types of trees with varying structures to illustrate this argument and put it in the main body, as this will add to the soundness of your methodology. Current A.1 is not straightforward enough, as I will discuss later below. Additionally, how can you apply norm loss for weighted hierarchies?
- Line 168/171: The format of i and e doesn’t match. Also i is defined twice, one in subscript of distance matrix, and one in the epoch number.
- Line 172: Why do you use Riemannian SGD if your encoder ResNet is fully Euclidean?
- Line 199: curvature c = -1. Curvature is not introduced in Section 2.2 where you hide all c terms in the operations. Although you set c = -1 by default, it’s better to show this term in equations. More importantly, curvature will affect the distance between points on the Poincare disk, which may have a significant impact on OOD detection. Please consider adding this in the experiment section for comparison.
- Line 228-229: “We also exclude … Euclidean”, Can you map your Poincare features to its tangent space with tangent map, so the features are on Euclidean space, and you can use Mahalanobis-distance based OOD score?
- Line 268: Can you include the details of “hierarchical out-of-distribution evaluations” as this is not common in OOD literature.
- Table 2: 89.1 => 89.10, 74.4 => 74.40 the number of digits should be consistent across tables. Compared to Table 1, why do you only use 5 OOD scores? Can you show the comparison of base and your method for each OOD dataset for Table 1 and Table 2?
- Figure 2: Which in-distribution dataset do you use to plot this figure? What about the comparisons of AUPR and n-AUROC for the same setting?
- Line 362-365 “Several … options”: since you initialized with [3]’s representation (line 176-177), it’s not surprising that your method is better than [3] because [3] is not designed for OOD detection.
- Line 369-371 “These values … standard classification”: Again [3] and HEC are not designed for classification. In my opinion, it is a method to visualize hierarchical data with lower distortion of input hierarchies. This is also the reason why comparing with [1] [4] [5] etc. is important and you can try to apply the Poincare Multinomial Logistic Regression layer on top of [1].
- Table 3: How do you compute the distortion metric and accuracy metric in Table 3? It seems PE has high AUPR and AUROC values, and can you explain this advantage (and disadvantage of your method)?
- Table 4: Why do you choose k = 300? In [6], k is 200 for CIFAR100, 50 for CIFAR10.
- Hierarchical Ablations: I’m a bit confused about this setting. Are hierarchical ood datasets really OOD data? Then in Table 6, you show that you can learn some hierarchical relationships on the OOD data, but is this going to hurt or improve OOD performance if you treat hierarchical held-out data as the OOD data? Why are FPR95/AUROC/AUPR metrics missing for this comparison?
- Table 6: What does “base” stand for in Table 6?
- Line 842-850:
  - I strongly recommend authors to move the motivations of your loss design to the main section for better readability.
  - Could you elaborate why ensuring equal distance to origin for nodes at the same level is beneficial for OOD? In hyperbolic embeddings, OOD samples are closer to the origin as you said, but how does training in distribution data with norm loss affect the location of OOD distribution? Shouldn’t we try to push all in-distribution samples away from the origin?
  - Please include details about the “toy tree example” you used to plot Figure 6.
  - Text in Figure 6 is too small.

[1] Ganea, Octavian, Gary Bécigneul, and Thomas Hofmann. "Hyperbolic neural networks." Advances in neural information processing systems 31 (2018).

[2] Zeng, Siqi, Remi Tachet des Combes, and Han Zhao. "Learning structured representations by embedding class hierarchy." The eleventh international conference on learning representations. 2023.

[3] Nickel, Maximillian, and Douwe Kiela. "Poincaré embeddings for learning hierarchical representations." Advances in neural information processing systems 30 (2017).

[4] van Spengler, Max, Erwin Berkhout, and Pascal Mettes. "Poincare resnet." Proceedings of the IEEE/CVF International Conference on Computer Vision. 2023.

[5] Yue, Yun, et al. "Hyperbolic contrastive learning." arXiv preprint arXiv:2302.01409 (2023).

[6] Sun, Yiyou, et al. "Out-of-distribution detection with deep nearest neighbors." International Conference on Machine Learning. PMLR, 2022.

---

> ### Author Response · Authors · 2024-11-23
> **Response to Reviewer ZNiu**
>
> We thank the reviewer for their feedback. Below, we have addressed the reviewer's comments regarding difference over existing works and additional analyses. We have also gone over all individual questions. We believe that the review and rebuttal strengthen the paper and we hope to have answered all questions adequately.
>
> **Novelty over existing works.** Indeed, [1] and [2] have previously highlighted the potential of hyperbolic or hierarchical embeddings for out-of-distribution detection. We take inspiration form such works and make following novel contributions:
>
> - The proposed balanced hyperbolic embeddings approximate tree distances more accurately than existing optimization approaches and can also serve as a standalone embedding tool.
> - We introduce generalizations of existing OOD detection scores to their hyperbolic variants and provide in-depth analyses to showcase its effectiveness in OOD detection across datasets.
>
> We have added clarifications in the paper to highlight these key distinctions and novel contributions over existing works.
>
> **Changes in section 5.** For better clarity, we added a short introduction in section 5 and updated the headings to be more descriptive.
>
> **Motivation for losses and how norm loss affects OOD detection.** We aim to learn an embedding with minimal distortion to preserve the hierarchical information in the labels. This is achieved by jointly optimizing the distortion (Equation 7) and norm loss (Equation 9). Optimizing Equation (7) directly minimizes distortion. The reviewer is right in pointing out that in we should try to push all in-distribution samples away from the origin. The ID samples are indeed away from the origin when the distance to leaf nodes (corresponding to the classes) from the balanced hyperbolic embeddings is minimized. Consequently, it means we need norm loss in Equation (9) ensures that all nodes belonging to the same granularity of a hierarchy are equidistant from the origin.
>
> **Comparison to hyperbolic methods.** We added Poincare embeddings and HEC for completeness, and didn't expect them to be better in classification performance, as also explained in lines 347-377. The AUROC and AUPR performance of Poincaré embeddings (PE) highlights the strong connection between hyperbolic learning and effective OOD detection. However, a downside of PE is its low classification performance. In contrast, our method achieves strong performance in both classification and OOD detection. We add suggested comparisons to other hyperbolic approaches below.
>
> |  | FPR@95 | AUROC | AUPR |
> | --- | --- | --- | --- |
> | HNN [1] | 64.53 | 73.79 | 60.46 |
> | PR [4] | 87.83 | 58.27 | 37.73 |
> | HCL [5] | 62.31 | 75.43 | 59.10 |
> | Ours | **49.46** | **82.43** | **70.41** |
>
> **k = 300 in Table 4.** We choose k=300 as to compare to ablations from CIDER [Ming et.al., 2023]  and PALM [Lu et.al., 2024] [2]. As suggested, we also show below the results for k=200.
>
> |  |  | k=200 |  |  | k=300 |  |
> | --- | --- | --- | --- | --- | --- | --- |
> |  | FPR | AUROC | n-AUROC | FPR | AUROC | n-AUROC |
> | CIDER | 43.98 | 85.52 | 76.63 | 43.24 | 86.18 | 75.43 |
> | PALM | 39.01 | 87.46 | 79.36 | 38.27 | 87.76 | 78.96 |
> | Ours | **35.66** | **89.15** | **79.07** | **35.83** | **89.45** | **78.50** |
>
> **Eq (11) and comparing Euclidean vs Hyperbolic logits.** Equation (11) ensures that embeddings of images are optimized to be closer to their class prototypes, $P_\mathbb{B}$. This results in tighter, more compact ID clusters.
> Both Euclidean and hyperbolic logits perform equally well in classifying ID samples, as shown in Fig. 1. However, hyperbolic logits are particularly advantageous for identifying OOD samples due to the exponential nature of distances in hyperbolic geometry.
> A key distinction of our method compared to hyperbolic OOD scoring approaches such as [1], which use distance hyperplane boundaries, or [Khrulkov et.al., 2020], which rely on sample norms to distinguish ID from OOD, is that our method differentiates samples based on their proximity to the prototypes. While [1] and [Khrulkov et.al., 2020] might misclassify OOD samples with high norms as ID, our approach leverages more nuanced information for better discrimination.
>
> **Line 157-159; Regarding [2] and CPCC loss.** [2] and our method both enforce the hierarchical knowledge of the tree into the feature space but in different ways. [2] enforces hierarchy during training by using CPCC loss as a regularizer to learn tree distances between class means. In contrast, our method first explicitly embeds the label hierarchy into hyperbolic space. Using the leaf nodes corresponding to classes as prototypes, we then train the model on the dataset. We add the discussion to related work section.

---

> > ### Author Response · Authors · 2024-11-23
> > **Continued response to Reviewer ZNiu**
> >
> > (contd.)
> >
> > **Mahalanobis distance scoring on tangent space.** It is possible to apply Mahalanobis distance in the tangent space. However, the features after applying the logarithmic map to the tangent space do not satisfy the assumption of Mahalanobis distance, which requires the representation to follow a multivariate Gaussian distribution. For this reason, we exclude this distance comparison in our work.
> >
> > **L2 distance between $d_G$ and $d_\mathbb{B}$.** To clarify, we cannot take L2 distance in hyperbolic space between distance measures. Our formulation ensures that the contribution of the distance difference is relative and not dominated by points with larger hyperbolic distances.
> >
> > **Line 172: Riemannian SGD.** We use Riemannian SGD in only Algorithm 1 to learn the label hierarchy in the hyperbolic space. Afterwards, the labels become fixed prototypes. We train a ResNet34 with these fixed targets, which means all learnable parameters are in Euclidean space for that stage, and so we use SGD (line 293-294).
> >
> > **Visualizing learned embeddings.** We generate trees with various structures to learn balanced hyperbolic embeddings to illustrate the benefit of losses. The visualizations are added to Appendix B.
> >
> > **Line 141-144, quality of hierarchy.** We assume a correct and known hierarchy is available. For ImageNet-100, we use a pruned WordNet hierarchy, and for CIFAR-100, we use the available superclasses hierarchy. We do not use any LLM-generated hierarchies, leaving this exploration for future work.
> >
> > **About hierarchical OOD and ablations.** Hierarchical OOD and related metrics are designed for scenarios where an OOD sample overlaps with ID data, as similarly defined in previous works [a], [Betterino et.al. 2020]. For instance, this could represent a new, unseen bird species in a biological dataset. In such cases, it is crucial to achieve both granularity—classifying the bird as OOD—and precision—identifying the closest related ID class.
> >
> > Table 5 presents the standard OOD performance metrics (FPR/AUROC/AUPR) in the hierarchical setting, while Table 6 highlights the metrics that evaluate hierarchical properties as defined in [Betterino et.al. 2020] and [Dengxiong et.al. 2024].
> >
> > We have clarified this setup and provided further details in Appendix C.
> >
> > [a] Linderman, et al., Fine-grain Inference on Out-of-Distribution Data with Hierarchical Classification. COLLAs 2023
> >
> >
> >
> > **Questions for better clarity.**
> > - Line 70: “distance to the class prototype”, prototype either can be class mean or in our method, a predetermined prototype to reflect hierarchy of label space.
> > - Distortion in Table 3 is measured using metric from [Sala et.al., 2018] (referenced in Table 7) and now updated in Table 3 accordingly.
> > - Line 199; curvature c=-1, we updated the main text to include curvature in the equations.
> > - ID dataset for Figure 2 is CIFAR-100, now updated in caption.
> > - Added ablations of curvature, dataset-wise OOD results, additional scoring functions from Table 2, AUPR and AUROC from Figure 2 to Appendix C.
> > - Rewrote Line 146, “equivalent edge distances” and Algorithm 1 We updated notations in Algorithm 1 to improve readability.
> > - base in Table 6 refers to a Euclidean ResNet trained on the CIFAR-split.
> > - Lines 842-850, we added a brief motivation of losses to the main text and expand on it in the Appendix A. Added more details for toy tree example and update the Figure 6.

---

> > > ### Comment · Reviewer_ZNiu · 2024-11-29
> > >
> > > I want to thank the authors for carefully reading my review for each question, and I really appreciate the authors for making a significant effort on the revised submission. However, I want to emphasize several key points why I will still not give a higher score given the updated submission. The following sections are approximately ordered by importance sequentially from top to bottom.
> > >
> > > ## Norm Loss for OOD detection?
> > >
> > > I am aware of the new visualizations and experiments in the Appendix for the discussion of the distortion loss and norm loss. Thanks for providing these experiments. However, since one of your main contributions is on OOD detection, as shown in your title and keyword, these make me believe that norm loss is helpful for OOD detection. The authors also mentioned it in lines 886-888.
> > >
> > > The only takeaway message that I get from the paper (Figure 6 + motivation of losses in Appendix + Norm Loss section in the main body) is that norm loss successfully makes the vertices with the same depth having the same distance to the origin. This is expected due to the definition of your norm loss and success application of the optimizer, and I have no doubts.
> > >
> > > But, in the introduction, and your motivation Figure 1 and from previous literature, hyperbolic embeddings typically tend to place OOD data around the origin, and ID data near the boundary. Then the introduction of Norm Loss can hurt OOD data performance. For in-distribution nodes with lower depth (near the root node), with norm loss, assume they have the same depth, they are all equidistant to the origin. But how close are they to the origin? They can be equidistantly close to the origin, overlapping with the OOD data region, hurting OOD data performance. Without norm loss, on the same depth, it can be the case where there are only a few points close to the origin, and more points are near the boundary, then these few outliers do not have a strong impact on the OOD detection. If you want to boost the advantage of hyperbolic embeddings, boundary collapse of InD data seems to be very beneficial to leave enough space for OOD data, but as UQQS mentioned, your norm loss addresses boundary collapse, which contradicts the natural advantage of hyperbolic embeddings on OOD detection.
> > >
> > > Therefore, I don’t see a convincing or intuitive connection between your Norm Loss and OOD detection. For example, some very important questions that you need to address are:
> > >
> > > * How close are OOD data to the origin? (Also asked by UQQS)
> > > * Where does norm loss place the inD nodes w.r.t the origin and why?
> > >
> > > In Figure 2, if my understanding is correct, the last two columns for each set of the experiments are your method without vs. with this norm loss. For AUROC, there is almost no difference between balanced and unbalanced versions. Given AUROC is usually a more stable metric than FPR95 and AUPR in OOD detection from my experience, I again want to ask what is the real motivation of norm loss. It seems the Reviewer hwZY also asked about this question as well, but unfortunately, the logic in your reply *“The ID samples are indeed away from the origin when the distance to leaf nodes (corresponding to the classes) from the balanced hyperbolic embeddings is minimized. Consequently, it means we need norm loss in Equation (9) ensures that all nodes belonging to the same granularity of a hierarchy are equidistant from the origin.”* still confuses me.
> > >
> > > **I want to emphasize again that this is a very fundamental problem that authors have to address. Since you focus on the OOD detection, your methodology should be well-motivated for this purpose.** Otherwise, it can be hard for future researchers to use, analyze, and extend your work, and it can be difficult to understand or trust your empirical findings.
> > >
> > > I am also trying to guess if norm loss is related to other major evaluation metrics in your paper. However, I do not think it is related to distortion: since you train the current distortion metric directly, distortion is worse with additional objective terms. Regarding in-distribution accuracy, I would expect you need some specialized theoretical tools for arguments about the generalization of hyperbolic geometry, which can be quite challenging.
> > >
> > > In summary, I would like to see a very convincing argument of why we need this norm loss from the author.

---

> ### Comment · Reviewer_ZNiu · 2024-11-29
>
> ## Novelty over existing works
>
> Although this might be somewhat subjective, I still want to give my own reason for why the novelty of this work is relatively limited. Let me copy the authors' response for the novelty statement in the rebuttal and elaborate:
>
> *“The proposed balanced hyperbolic embeddings approximate tree distances more accurately than existing optimization approaches and can also serve as a standalone embedding tool.”*
>
> After you clarified your distortion metrics for evaluation and training, this novelty argument is not very strong. First, your distortion metric is borrowed from [Sala et.al., 2018], which is a recent work under a similar context. [Sala et.al., 2018] proposed this metric under the context of learning hyperbolic embedding, and authors directly use almost the same formulation as their training objectives. Besides, when you say “approximate tree distances more accurately,” I do not think this is a fair comparison with other embedding methods. If you directly optimize the distortion metric, as long as the optimizer works well, it is hard to achieve anything better than direct optimization.
>
> *“We introduce generalizations of existing OOD detection scores to their hyperbolic variants and provide in-depth analyses to showcase its effectiveness in OOD detection across datasets.”*
>
> Generalizations of existing OOD scores are a very simple extension if you learn embeddings in hyperbolic spaces. However, I am not sure if this amount of extension is enough novelty for ICLR. For the in-depth analysis, I agree authors provide extensive experiments, but the authors mainly focus on describing the numbers in the tables. Since authors include many metrics, datasets, baselines, and settings in the experiment section, it is important to carefully interpret both performance gain and performance loss/outliers for some set of experiments, so researchers can use your work more effectively in the future.
>
> ## Hierarchy for OOD detection?
>
> The paper claimed that OOD detection will benefit from both hyperbolic embeddings and hierarchical information. I don’t have too many concerns about the hyperbolic embeddings, but for the hierarchical information, first, it relies on the quality of hierarchy as you mentioned in the limitation. However, in practice in a more general OOD setting, such kind of ground truth tree information is usually unavailable, so I cannot verify if your method will generalize well other than CIFAR100, ImageNet, etc. Second, I would imagine, even if you have access to the ground truth hierarchy, it may hurt OOD detection performance when the inD and OOD are too near. If they share some parts of the ground truth hierarchy, introducing tree information can cause confusion in the OOD detection. This problem becomes even more challenging when we do not have access to the ground truth tree for the datasets. So the benefit of using hierarchy for OOD detection, although mentioned in some previous literature, is not very strong either.
>
> Furthermore, in the current submission, it is hard to see how much the OOD gain is from hierarchical information and how much is from hyperbolic embeddings. If the Euclidean baseline is just standard ResNet trained with cross-entropy, this baseline seems to be too weak for comparison. It is more reasonable to use the Euclidean version of your distortion loss (by replacing $d_\mathbb{B}$ with Euclidean distances) + norm loss for OOD detection, to prove the advantage of hyperbolic embeddings. And this will also help you show how much gain you get from including hierarchical information by comparing Euclidean hierarchical embedding with Euclidean + cross entropy embedding. If you are worried that Euclidean space cannot embed the tree very well, then another necessary baseline is just training with hyperbolic cross-entropy in Equation 11, and use this to compare with your final method. Some of the results are now included in Figure 2, but many OOD metrics are still missing.
>
>
> If I have more questions about some details of the paper, I will add them later and ask for explanations if necessary. But personally, these concerns mentioned above are more fundamental weaknesses of this work, so I want to summarize them here and I am open to further discussion. Besides, if the authors feel some mentioned experiments require a huge amount of effort and find it challenging to finish all during the remaining rebuttal time, feel free to point them out.

---

> > ### Author Response · Authors · 2024-12-01
> > **Response to additional questions from Reviewer ZNiu**
> >
> > We thank the reviewer for their feedback and appreciate the opportunity to clarify further questions. Below, we address each of your concerns, and will add minor requested changes (Fig 2, other OOD metrics) to the final version. We hope to have answered all questions adequately.
> >
> > **Norm loss for OOD detection.** The norm loss ensures that within a hierarchy all nodes belonging to a same level maintain a similar distance from the origin. By balancing the norms across the hierarchy, norm loss prevents imbalances that could otherwise lead to subtrees having disproportionately lower norms. This is important, because without this loss, smaller subtrees are placed closer to the origin (where OOD samples commonly occur), which leads to less uniform probability distributions for OOD samples and makes it harder to discriminate ID from OOD samples.
> >
> > *Where does norm loss place ID nodes w.r.t  This is origin?* In our training framework, all in-distribution (ID) classes are designated as leaf nodes of the hierarchy. Consequently, norm loss guarantees that all leaf nodes maintain an equal distance from the origin. Because leaf nodes in a hierarchy are positioned farther from the the root node, ID classes are consistently placed near the boundary, leveraging the natural advantage of hyperbolic embeddings. This relationship is also illustrated in Appendix B, Figure 7 (right), where the leaf nodes (the final level of the hierarchy) have the highest norms, while ancestor nodes that are necessary to establish a hierarchical relationship have lower norms.
> >
> > *How close are OOD data to the origin?* As illustrated in Appendix C, Figure 8, OOD data consistently reside closer to the origin compared to ID data.
> >
> >
> >
> > **Novelty over existing works.** Our novelty encompasses the combination of both things below:
> >
> > (1) While distortion, a standard graph metric also used in [Sala et.al.,] is now mostly used for evaluation only, we do not claim distortion itself as our contribution. Our contribution is Algorithm 1, which combines distortion and norm loss to learn the hierarchy embeddings in the hyperbolic space.
> >
> > (2) We provide in-depth analysis of across multiple settings—standard OOD and hierarchical OOD—and embedding spaces—comparison to hyperbolic embeddings and ablations w.r.t euclidean and non-hierarchical hyperbolic embeddings.
> >
> > **Hierarchy for OOD Detection.** As mentioned in our response to reviewer UQQS, our work relies on a predefined hierarchy, which we have now included in the limitations. Regarding OOD detection when ID and OOD data are closely related, we demonstrate that in hierarchical generalizations—specifically within the CIFAR50/50 split, where ID and OOD classes share common ancestors in the ground truth tree—our method is beneficial. Our approach not only improves OOD detection performance, as shown in Table 5, but also enhances the identification of the closest ID class or ancestor, as detailed in Table 6.
> >
> > **Disentangling gain from hyperbolic embeddings and hierarchy.**
> > Thank you for the great suggestion. We have implemented a hyperbolic cross entropy loss as an additional baseline to show the gain from using hyperbolic space vs using additional hierarchy information. Euclidean version of distortion loss might not be feasible in this short time frame, but we plan to add it in the final version for completeness.
> >
> >
> > |  |  | FPR95 |  |  | AUROC |  |
> > |:---:|:---:|:---:|:---:|:---:|:---:|:---:|
> > |  | Euc CE (base) | Hyp CE | Ours | Euc CE (base) | Hyp CE | Ours |
> > | MSP | 58.24 | 58.08  | **49.46** | 77.05 | 78.34  | **82.43** |
> > | TempScaling | 54.29 | 58.08 | **48.61** | 78.18 | 78.34 | **83.02** |
> > | Odin | 60.96 | 62.50 | **49.45** | 76.63 | 76.52 | **82.96** |
> > | Gram | 83.33 | 69.26 | **57.78** | 62.31 | 73.23 | **76.84** |
> > | Energy | 58.47 | 59.81 | **55.41** | 77.65 | 78.22 | **81.74** |
> > | KNN | 47.95 | 55.25 | **44.00** | 83.29 | 78.40  | **85.50** |
> > | DICE | 64.61 | 75.24 | **53.56** | 74.35 | 70.97  | **82.80** |
> > | RankFeat | 73.03 | 72.05 | **52.88** | 68.98 | 69.25  | **77.97** |
> > | ASH | 67.48 | 82.53 | **53.48** | 76.88 | 67.83 | **79.15** |
> > | SHE | 77.07 | 74.47 | **56.05** | 67.09 | 75.60  | **81.91** |
> > | GEN | 54.66 | 58.77 | **47.20** | 79.21 | 78.53  | **83.80** |
> > | NNGuide | 65.44 | 95.88 | **50.01** | 76.37 | 35.05 | **82.75** |
> > | SCALE | 57.65 | 73.17 | **51.33** | 79.68 | 76.20 | **82.05** |

---

> ### Comment · Reviewer_ZNiu · 2024-12-01
>
> Thanks for your reply.
>
> **Norm Loss** Again your mathematical expression of norm loss cannot guarantee the position of inD data and OOD data for their distances to the origin. Therefore, I still do not understand why it can work as now it works magically. I believe my counterexamples provided before, no matter where my vertex is with respect to the origin, as long as all vertices are at the same depth/level, my norm loss will be small. It can be contradictory to the hyperbolic embeddings for OOD detection’s major advantage. Even if all inD nodes are leaf nodes, again, their distances to the root node/level is based on the ground truth tree, but they can be either large or just 1.
>
> Besides, I don’t think all leaf nodes’ distance to root is the same for ImageNet100 is the same, based your figure 7 in the Appendix. If they are indeed placed equidistant to the origin, probably some of your implementation is wrong.
>
> **Novelty** Because you claim your contribution to the combination of distortion and norm loss, while the norm loss’s motivation is still questionable, this novelty statement is not very strong in my point of view. This is the reason why I keep asking clarifying questions for the norm loss.
>
> **Hierarchy for OOD detection** For the nearOOD performance on CIFAR-OSR split, the same concern for the weak baseline applies, and you need to provide more comprehensive analysis (with several controlled splits sharing the same hierarchy, stronger baselines, more inD datasets, more OOD metrics etc.) to illustrate the point that your method even works for nearOOD methods. I didn’t expect the authors to address them in the rebuttal time but this will be a good thing to add.

---

> > ### Author Response · Authors · 2024-12-03
> > **Clarifications**
> >
> > Thank you for getting back to us. Based on the comments, we want to make a few clarifications that hopefully make our approach better to understand:
> >
> > **Role of norm loss for OOD.**
> > We apologize for the confusion, the norm loss is *not* used when training networks and hence does not directly force upon ID/OOD samples. The norm loss is only used when constructing the hyperbolic embeddings of the hierarchies. The idea behind the norm loss is to enforce similar norms for nodes at the same level of abstraction. We find that existing approaches ignore the problem of imbalance between subtrees. When training a network, ID samples end up near the boundary of hyperbolic space. During testing, OOD samples tend to gravitate towards the origin. We show this in Figures 4 and 8. Without a norm loss, shallow subtrees also gravitate towards the origin, leading to unwanted biases. Hence our improvements in Table Figures 2,7 and 10.
> >
> > **Dealing with multiple hierarchical levels.**
> > We want to clarify that norm loss is applied to same levels of abstraction, i.e., we group nodes together starting with the leaf nodes as the first group. We will add this detail to the paper.
> >
> > **Novelty.**
> > We hope that the clarifications regarding the norm loss help to solidify the approach.
> >
> > **Hierarchical generalization experiments.**
> > We have added the hyperbolic cross-entropy baseline to the main Tables, where our method remains best. We will do the same for the hierarchical experiments.

---

### Official Review · Reviewer_UQQS · 2024-11-01

**Soundness:** 3
**Presentation:** 4
**Contribution:** 3
**Rating:** 8
**Confidence:** 4

**Summary:**

This work first identifies that hyperbolic embeddings are stronger at differentiating between in and out of distribution than euclidean methods due to the inherent distance / hierarchy property of the space. The authors identified that OOD points will lie closer to origin and as such have a more uniform distribution to classification prototypes placed at the boundary. Following this observation, the authors present a new distortion based loss function to match embedding targets to a known hierarchy, and balances hierarchy levels to ensure that correct hierarchical depths are preserved. This method is then applied to  to a variety of OOD scoring functions to demonstrate wide applicability. The results show that the proposed method outperforms Euclidean and hyperspherical approaches over a variety of benchmarks and OOD scores and visualisation mostly support their hypothesis. All implementations and empirical evaluations are described in full detail for replication.

**Strengths:**

**Structure and Clarity:**
- The problem statement is very clear, hypothesis is well reasoned, and the justification well described.
- Preliminaries are concise yet highly informative, and add little bloat to the work while providing the reader with the necessary understanding for both OOD and hyperbolic learning.
- Results are presented clearly, with descriptive figures, graphs and accompanying written discussions.

**Method hypothesis, findings, and rationale:**
- The method is very well motivated by observations and key findings of hyperbolic deep learning, where the properties of hyperbolic distances wrt distance from the origin are leveraged.
- Balancing all learnt embeddings to match the known tree like hierarchy is a nice addition, while it has its limitations as addressed below, should enforce correct hierarchical structure in representation space.
- The proposed method is simple in each of its components, allowing for simple and neat incorporation into a variety of existing settings, providing good impact and insight, valuable to the community.

**Reproducibility:**
- All details are presented for full reproduction in the text including optimisation, hyperparameterisation, datasets, and architectural settings, with the addition of algorithm of the method further adding clarification.

**Experimental results:**
- The performance improvements of the proposed method in comparison to Euclidean approaches are significant across almost all OOD scoring methods, providing clear justification for its future use.
- A variety of experimental settings are evaluated, while more could always be added, that present are enough to provide confidence on the generalisation of the method.

**Ablations:**
- The presented ablations do a good job of analysing each proposed component and justify empirically its value in the proposed system in addition to being visually very interpretable.
- The addition of comparisons to hyperspherical work appropriately evaluates this work in line with existing and preferred methods, which further provides confidence of the findings.

**Weaknesses:**

**Learning a well defined hierarchy:**
- One assumption you make is that the hyperbolic method does indeed learn a strong hierarchy. However, this work does not demonstrate empirically or theoretically that the hierarchy is in fact learnt. The distortion loss should help enforce this structure, however, empirical analysis would be a great addition.
- Following from the prior, it is well known that in hyperbolic space embeddings can “collapse” to the boundary of the ball, and hence no hierarchy is learnt. Do you provide evidence that this does not happen or that the hierarchy is indeed present? While the norm loss should help address this, it would good to see how this holds as prior attempts to regularise hyperbolic embeddings via norm result in clipping like effects. While you show oil ablation comparisons to clipped embeddings, directly

**Assumption of a known and correct hierarchy:**
- Less of a weakness and more of a limitation, the method assumes access to a well defined and known hierarchy. While in the demonstrated case this is know, in most settings this is not the case. Therefore the generalisation of this method is somewhat limited
- As mentioned above, while an understandable and reasonable limitation, this needs to be clarified and addressed explicitly as part of the limitations of the work.

**Distance of OOD:**
- Figure 3 measures the distance between embedding of each distribution, however to more appropriately support your hypothesis of OOD points being lower shallower nodes in the tree, a more appropriate analysis would be on the norm of the embeddings from the origin.
- As can be seen from figure 4, the points can achieve high distance but lie very deep in the tree given the increase in hyperbolic pairwise distance wrt to distance of the points from the origin. Therefore, for the most part your hypothesis holds, unknown points lie closer to the origin. However, this is not always the case.
- Furthermore, from the prior point, do you have any intuition how so much of the OOD points are taking up space close to the boundary? If you assume a relatively uniform distribution of embeddings of the hierarchy during training of ID points then figure 4 embeddings for OOD are lying in the hierarchy of an ID point.

**Minor:**
- Figures 1 a is informative and descriptive for the problem setting and help visually demonstrate the properties of hyperbolic space. However, 1 b is less informative and is not clear if this is an illustration or real embedding positions? If simply an illustration, it does not add a great deal to the narrative or work. I assume the latter, but would argue that figure 1 a is a clear selling point.
- Following the prior point, it would be good if the figures are slightly larger and the points made more distinguishable.

**Questions:**

1. How does the method perform if the OOD same is highly related to the semantic concepts captured in the learnt hierarchy. This method assumes that the OOD sample is semantically very different from the training dat
2. Does the method generalise well to other curvatures, given it has been shown that most visual embeddings are not fully hyperbolic, and thus different curvatures may be optimal for different datasets.
- Additional questions are asked for improve clarity to the authors in the weaknesses section.

---

> ### Author Response · Authors · 2024-11-23
> **Response to Reviewer UQQS**
>
> We thank the reviewer for their detailed feedback, we address the question below and fix minor mistakes in the paper directly and update Figure 1 to be bigger.
>
> **Learning a well defined hierarchy**. We agree that distortion loss helps enforce the hierarchical structure. The reviewer is also correct in noting that the combination of distortion and norm loss helps prevent embeddings from collapsing. Measuring distortion of the learned graph (ex., distortion in Table 3, 7) shows that the hierarchical distances are preserved in the hyperbolic embeddings. To Appendix B, we add a plot of pairwise distances of each nodes which confirms that the hierarchy is indeed preserved. Additionally, we add a plot average norm for all the levels of hierarchy for CIFAR-100 and ImageNet-100.
>
> **Assumption of a known and correct hierarchy.** We agree that the method assumes access to a known and correct hierarchy. Updated the limitations to include this.
>
> **Distance of OOD.** The observations made by the reviewer in Figure 4 are correct, we add two clarifications:
>
> - On average OOD samples are closer to origin. As suggested, a plot showing the distribution of norms is added to the Appendix C.
> - OOD samples can have a high norm without necessarily being close to the prototypes.
>
>
> **If OOD is highly related to the semantic concepts.** Using hierarchical prototypes makes it easier to handle OOD examples that are closely related to in-distribution data, as shown in hierarchical ablations (CIFAR-OOD-Split). It also leaves room to build on existing hierarchical datasets from other research or come up with new methods, making it a flexible and practical approach for future improvements.
>
> **Generalization to other curvatures.** The method can generalize well to curvatures which can be dataset dependant. We keep the curvature fixed to -1 for our main experiments. We will add an ablation of curvature to Appendix C.

---

### Official Review · Reviewer_rQmy · 2024-11-04

**Soundness:** 3
**Presentation:** 3
**Contribution:** 3
**Rating:** 6
**Confidence:** 3

**Summary:**

This paper proposes hierarchical hyperbolic embedding for out-of-distribution detection. Specifically, it introduces a balanced hyperbolic embedding that maintains a similar distance between any two nodes. The learned hyperbolic embedding demonstrates superior performance compared to existing OOD approaches across various benchmarks and scoring functions, and its effectiveness is validated through numerous ablation studies.

**Strengths:**

The hierarchical hyperbolic embedding shows promise for OOD detection, which is an interesting idea.

The effectiveness of the method is demonstrated through comparisons with various benchmarks and ablation studies.

Overall, the paper is clear.

**Weaknesses:**

1) The motivation of “hierarchical” hyperbolic embedding for OOD detection is somewhat unclear.
Could you please clarify the motivation behind using hierarchical relationships for hyperbolic embedding in OOD detection? Although it is well-known that hyperbolic embeddings can effectively represent distances in hierarchical graphs, it’s a little confusing about how this specifically benefits OOD detection.

2) It would be helpful to include some recent related works on hierarchical hyperbolic embedding [1, 2].

[1] Unsupervised Hyperbolic Metric Learning, CVPR, 2021
[2] HIER: Metric Learning Beyond Class Labels via Hierarchical Regularization, CVPR, 2023

**Questions:**

Related to W1, could you please clarify the motivation behind using hierarchical relationships for hyperbolic embedding in OOD detection? Although it is well-known that hyperbolic embeddings can effectively represent distances in hierarchical graphs, it’s a little confusing about how this specifically benefits OOD detection.

---

> ### Author Response · Authors · 2024-11-23
> **Response to Reviewer rQmy**
>
> We thank the reviewer for their positive feedback and clarify our motivation below.
>
> **Motivation of hierarchical embeddings for Hyperbolic OOD.**  Previous research has demonstrated that hyperbolic spaces are highly effective for OOD detection, as briefly shown in Fig. 1 and further explored in the ablation study in Fig. 2 (non-hierarchical hyperbolic). This establishes a strong foundation for our approach to OOD detection using hyperbolic spaces. This forms a strong basis for our approach to OOD detection using hyperbolic spaces. Using hierarchical relationships for OOD adds two benefits:
>
> 1. Incorporating hierarchical relationships introduces meaningful structure to the embedding space, where the distances between class prototypes reflect their hierarchical relationships.  This makes it easier for OOD detection methods to differentiate between samples from closely related classes and those that are genuinely out-of-distribution, particularly in challenging scenarios where OOD samples resemble in-distribution (ID) classes.
> 2. Learning with hierarchical prototypes enables our method to generalize to both distance-based and logit based scoring functions.
>
> **Adding recent related works.** The paper is updated to include citations to the related works, thanks for pointing them out.

---

### Official Review · Reviewer_hwZY · 2024-11-04

**Soundness:** 2
**Presentation:** 2
**Contribution:** 2
**Rating:** 5
**Confidence:** 5

**Summary:**

## Summary

The paper proposes Balanced Hyperbolic Learning for improved out-of-distribution detection by utilizing the hierarchical label information and learning more discriminative representations between ID and OOD samples. This is done in a two-step hierarchical hyperbolic embedding optimization process, where the first step involves a combination of distortion loss and norm based loss to learn the hyperbolic class prototypes from the label hierarchy. Then, the second step involves obtaining the hyperbolic representation for each image by optimizing a hyperbolic distance-based cross entropy loss. Experimental results are provided on two ID datasets for a suite of near and far OOD datasets, and involves comparison with well known OOD detection techniques from literature.

---

**Strengths:**

## Strengths

1. The paper is generally well organized.

2. Connections between hyperbolic representation learning and out-of-distribution detection have been evidenced by prior work and new contributions at this intersection are interesting and valuable to the community.

3. The authors provide a good coverage of related works in out-of-distribution detection, hyperbolic embeddings of images, and hyperbolic learning of hierarchies, to place their proposed method in the right context.

4. The authors propose generalizations of existing OOD detection scores to hyperbolic variants which are simple and easy to adapt.

5. Hyperbolic distance based loss function during optimization as in (11) is intuitive and naturally suitable for learning discriminative representations for distance-based OOD detection

---

**Weaknesses:**

## Weaknesses

1. Some of the claims made by the authors are not substantiated by prior work or experimental evidence, for instance L077 “existing hyperbolic embedding methods are biased towards deeper and wider sub-trees, with smaller sub-trees pushed towards the origin” - how do the authors define the bias and how is this verified experimentally?

2. Key details of the setup and specific experiments are missing, making it difficult to accurately evaluate the comparison with prior methods and impact of the work (see list of questions below for missing details)


3. The goal and setup of the OOD detection task is to improve OOD detection performance  while maintaining the ID accuracy, however the authors do not report the ID accuracy as compared to the baseline methods for either the CIFAR100 or the ImageNet100 datasets. The ID accuracy is reported for only a subset of methods for CIFAR100 in Table 3, and the reported ID accuracy for the proposed method seems to be lower than the accuracies reported by other methods, for instance CIFAR 100 ID acc is 73.4 whereas the CIDER paper reports ID acc of 75.35 on CIFAR100 with ResNet34 (page 19).


4. The design of the loss function in Eq. 7 can potentially lead to scale mismatch issues due to the difference in the source and nature of the hyperbolic distance $d_B$ and graph distance $d_G$. $d_B$ grows non-linearly as the points approach the boundary and can dwarf $d_G$ which remains typically bounded and grows linearly w.r.t the edge counts, especially in smaller graphs. This can cause mis-alignment where large $d_G$ does not map proportionately to large $d_B$, did the authors investigate the scales of these terms and the overall loss trends?


5. Some pivotal connections to prior works are not fully acknowledged or discussed, for instance the primary hypothesis of this paper from Fig 1(b) and L082 - “OOD samples lie between ID clusters and the origin” is similar to an identical hypothesis proposed in CIDER (Figure 2, page 4 of the CIDER paper), albeit in hyperspherical embeddings, the key idea remains the same.

---

**Questions:**

## Questions

1. What is the loss function and experimental setup used for all the baseline (non-hyperbolic) experiments as reported in Tables 1 and 2? L250-253 mention using the “same features as in existing works for the most direct comparison to Euclidean-trained counterparts” which indicates that the methods and losses from original works are used, whereas L310-315 mentions “for the baseline and ours, we use the exact same backbone and training procedure” which is confusing since the loss function for the two settings (original and proposed balanced hyperbolic) are expected to be different. Additionally, did the authors experiment with the Supervised Contrastive Loss [1] for the baseline Euclidean setting? Empirically much better results are reported using the euclidean SupCon loss in SSD [2] and KNN (Sun et. al, 2022) as compared to the cross entropy loss.


2. How is the distortion measured in Table 3?


3. How do the learnt representations and hierarchies differ when the proposed method is used without the norm-balancing loss? While this is observed via the OOD detection performance in 349-360, how does it affect the learning intuitively?

4. Since the initialization of the hyperbolic prototypes is dependent on another technique, how do the authors see their method generalizing to other models of hyperbolic geometry?


5. Have the authors visualized the learnt hierarchies from the hyperbolic embeddings to verify if the underlying hierarchical relationships are indeed accurately encoded in the hyperbolic space using the two-step optimization process?

---

## Suggestions on improving presentation:

1. The description in Section 3.2 (154-161) does not mention any details about the initializations of the prototypes, this description can be moved up from the next section to provide more context

2. “distortion” is mentioned in the introduction and Section 3.1 several times without any references or description, this should be included

3. The Algorithm 1 on page 4 should include an “Output” marker and corresponding notations to denote the expected result from the optimization process

4. Some minor typographical fixes - 157 “corresponding the the n graph..” -> “corresponding to the ..” , 183 “should have a the same..” -> “should have the same norm”, 417 “we have the use backbone for all methods” -> “we use the same backbone for ..”, etc.

---


## References:

[1] Supervised Contrastive Learning, Khosla et. al, NeurIPS ‘20

[2] SSD: A Unified Framework for Self-Supervised Outlier Detection, Sehwag et. al, ICLR ‘21

---

## Post Rebuttal

I thank the authors for the rebuttal. Having gone through their responses, their are several issues raised in my original review about the completeness and presentation of the results that are still not addressed during the rebuttal, specifically the discrepancy in the in-distribution accuracies and reported results for only a subset of the settings, and details about the choice of experimental setup and it's implications. Therefore, I would like to maintain my score.

---

> ### Author Response · Authors · 2024-11-23
> **Response to Reviewer hwZY**
>
> We thank the reviewer for suggesting related works and experiments, as well as visualizations to enhance the paper. We answer the questions below and correct minor mistakes directly in the paper.
>
> **On bias towards deeper and wider subtrees.** The loss function in Equation (7) leads to embeddings for which the hyperbolic distances between the embedded vertices closely resemble the graph distances. Hierarchy imbalance is caused by the uneven distribution of labeled nodes which can cause uneven norms corresponding to the imbalance in works that don't explcitly correct for it [Nickel et.al, 2017, Ganea et.al. 2018].  We generate synthetic hierarchies for essential verification and compare with the hierarchies generated by previous works. We will add the visualization to Appendix A.
>
> **ID performance.** The reported accuracy for our hyperbolic model is comparable to baseline CE models, while offering significant advantages in OOD detection. When training CIFAR-100 under a setting similar to CIDER (75.35), our method achieves nearly identical ID performance (75.24).
>
> **Possible scale match issue in loss function.** The division by $d_G$ in Equation (7) actually prevents any distant pairs contributing large values. The non-linear growth of $d_\mathbb{B}$ is with respect to the Euclidean representation that is being used to model hyperbolic space, i.e. the Poincaré ball, which only influences the embedding through possible numerical errors.
>
> **Connection to previous papers.** We agree and have added a reference to CIDER in the paper accordingly. In CIDER, OOD samples lie only between ID clusters because the embeddings are normalized onto a hypersphere. In contrast, in our method, OOD samples lie both between ID clusters, and between ID clusters and the origin, as hyperbolic space allows embeddings to exist anywhere within the space.
>
> **More details of experiment setup.** Loss function for baseline experiments is cross-entropy loss, experimental setup is similar to OpenOOD benchmark. L250-253 is updated  for clarity. L310-315, the models differ in how logits are computed but the loss is still cross entropy loss (Eq 11), so training hyperparameters remain the same. Distortion metric is defined in [Sala et.al., 2018] as referenced in Table 7 and now updated for Table 3  accordingly. We update necessary details in the paper and in Apendix C.
>
> **Using SupCon loss instead of CE loss.** Methods using SupCon loss evaluate OOD using only distance-based functions (e.g., Mahalanobis score in SSD+ and KNN score in KNN+). In contrast, our method generalizes to both distance-based and logit-based scoring functions. Therefore, the baseline Euclidean setting uses CE loss to enable comparison across the scoring functions. For completeness, we have now included comparisons to SSD+ and KNN+ (in the context of table 4) to Appendix C.
>
> | **Method** | **FPR@95 ↓** | **AUROC ↑** | **n-AUROC ↑** |
> | --- | --- | --- | --- |
> | SSD+ | 57.13 | 80.27 | 77.13 |
> | KNN+ | 54.46 | 79.29 | 77.81 |
> | CIDER | 43.24 | 86.18 | 75.43 |
> | PALM | 38.27 | 87.76 | **78.96** |
> | Ours | **35.83** | **89.45** | 78.50 |
>
>
> **How does norm loss affect the learning?** By adding Norm loss, all ID prototypes in the label hierarchy are positioned at the same distance from the origin in hyperbolic space, even for imbalanced hierarchies. These prototypes are then fixed for the next stage. When training a ResNet-34, this ensures that all ID embeddings are farther from the origin, allowing OOD samples to map closer to the origin or between the prototypes.
>
> **Can the method generalize to other hyperbolic models?** We first randomly initialize the model and learn Poincaré embeddings for 100 epochs, after which we continue training with our distortion loss. We now clarified this in the setup as well. Since a distance function is well-defined for various models of hyperbolic geometry, making it possible to generalize this formulation across different representations of hyperbolic space. Alternatively, prototypes can be learned in the Poincaré ball and mapped isometrically to other models.
>
> **Visualizing learnt hierarchies.** Measuring distortion of the graph (ex. in Table 3 and 7) shows that the hierarchical distances are preserved in the hyperbolic embeddings. In the Appendix B, we also visualize the pairwise distances of each nodes in the hierarchy to show the hierarchical structure that emerges.

---

### Author Response · Authors · 2024-11-23
**Summary of rebuttal**

We thank all reviewers for their appreciation of our work: novel and valuable (*hwZY, UQQS*), yet simple and widely applicable (*hwZY, UQQS, ZNiu*), with comprehensive experiments and ablations (*rQmy, UQQS*).

We also thank the reviewers for the constructive suggestions. We have revised the main text for better readability (now highlighted in blue) and added additional suggested experiments and visualizations to the appendix. The PDF will be  uploaded shortly.

We are running the ablations on curvature (*ZNiU*) and validating bias towards deeper and wider subtrees (*hwZY*) and the results will be added to the appendix.

---

> ### Author Response · Authors · 2024-11-28
> **Updated suggested changes and analyses in the PDF.**
>
> We again thank the reviewers for their thoughtful feedback, which has helped us further refine the paper. We have incorporated your suggestions and updated the PDF, with all changes highlighted in blue. Below is a summary of the revisions:
>
> 1. **Improved Clarity and Readability**: We refined the main text to enhance its clarity, readability, and completeness. Specifically, we increased the size of Figure 1, clarified equivalent edge distances in line 149, improved the description of Algorithm 1, added a citation for distortion in Table 3, and expanded the discussion on scoring function comparisons in lines 233–255. Additionally, we ensured consistent decimal formatting across all tables.
> 2. **Structural Improvements**: To improve structure and presentation, we added a brief overview paragraph before discussing the experiments in Section 5 and revised the headings for the hyperbolic comparisons and hierarchical ablations. We expanded the details of the hierarchical generalization in Section 5, with further elaboration provided in Appendix C. To accommodate these updates, we moved portions of the Related Work section to Appendix D.
> 3. **Ablations and Analyses**:  We included multiple new ablations and analyses in Appendix C to strengthen our findings. These additions include expanded results on ImageNet-100, dataset-wise results on CIFAR-100, and ablations focusing on AUPR, AUROC (Figure 2), and curvature.
> 4. **Additional Discussions and Visualizations**: : Appendix A includes a discussion of biases toward deeper and wider subtrees. Appendix B provides visualizations of the learned hierarchies. Appendix C now includes detailed experimental setups for the Euclidean baseline and analyses on ID/OOD embedding norms. We also added further details on Figure 6 to clarify the motivation for our proposed losses.
>
> We hope these revisions address the feedback and further enhance the quality and clarity of the paper. We welcome any additional questions or suggestions.

---

### Meta-Review · Area_Chair_PTqM · 2024-12-20

**Metareview:**

The paper proposes Balanced Hyperbolic Learning for improved out-of-distribution detection by utilizing the hierarchical label information. It has received constructive reviews, detailed comments, and also mixed ratings. On one hand, the reviewers appreciate the clear structure and organization of the paper, as well as the significant improvements in comparison to Euclidean embeddings in OOD detection. On the other hand, significant concerns have also been raised regarding the technical details and experimental setups that are missing.

Authors and reviewers engaged in extensive discussion during the rebuttal period, but nevertheless, the response was not able to fully address the reviewers' questions. In particular, the use of norm loss is not well supported in principle. Empirically, reviewers also had questions on the discrepancy in the IID accuracies and reported results for only a subset of the settings.

The authors are encouraged to incorporate the reviewers' comments to further strengthen the work when preparing for the next iteration of the paper.

**Additional Comments On Reviewer Discussion:**

The reviews are mixed. Both two expert reviews share some common concerns of the paper that have not been properly addressed during the discussion session. The decision was made largely based on the expert reviews.

---

### Decision · Program_Chairs · 2025-01-22

Reject